A phylogenetic taxonomy of the Cyrtodactylus peguensis group (Reptilia: Squamata: Gekkonidae) with descriptions of two new species from Myanmar

Grismer L. Lee 1 lgrismer@lasierra.edu
http://orcid.org/0000-0001-8406-0171 Wood Perry L. Jr 2
Quah Evan S.H. 3
Murdoch Matthew L. 4
Grismer Marta S. 1
Herr Mark W. 2
Espinoza Robert E. 5
http://orcid.org/0000-0001-5338-0658 Brown Rafe M. 2
Lin Aung 6
1 Herpetology Laboratory, Department of Biology, La Sierra University , Riverside, CA , USA
2 Department of Ecology and Evolutionary Biology and Biodiversity Institute, University of Kansas , Lawrence, KS , USA
3 School of Biological Sciences, Universiti Sains Malaysia , Penang, Penang , Malaysia
4 Department of Biology, Villanova University , Villanova, PA , USA
5 Department of Biology, California State University, Northridge , Northridge, CA , USA
6 Department of Biology, Fauna and Flora International , Yangon , Myanmar
Crandall Keith
Electronic publication date: 2018 Sep 19
Publication date: 2018
Volume: 6
Electronic Location ID: e5575
Received 2018 Apr 12; Accepted 2018 Aug 10
Copyright: © 2018 Grismer et al.
Copyright year: 2018
Copyright holder: Grismer et al.
License: This is an open access article distributed under the terms of the Creative Commons Attribution License, which permits unrestricted use, distribution, reproduction and adaptation in any medium and for any purpose provided that it is properly attributed. For attribution, the original author(s), title, publication source (PeerJ) and either DOI or URL of the article must be cited.
License URL: https://creativecommons.org/licenses/by/4.0/

Keywords: Molecular phylogenetics, Herpetology, Zoology, Taxonomy

Funding: College of Arts and Sciences of La Sierra University and Fauna & Flora International Life Science Museum at Brigham Young University NSF grant EF-1241885 NSF Graduate Research Fellowship College of Science and Mathematics at California State University, Northridge L. Lee Grismer was supported by the College of Arts and Sciences of La Sierra University and Fauna & Flora International. Fieldwork for Perry L. Wood Jr was supported the Monte L. Bean Life Science Museum at Brigham Young University and generation of molecular data was supported by the NSF grant EF-1241885 issued to Jack W. Sites. Participation of Mark W. Herr was supported in part by an NSF Graduate Research Fellowship. Robert E. Espinoza was supported by the College of Science and Mathematics at California State University, Northridge. There was no additional external funding received for this study. The funders had no role in study design, data collection and analysis, decision to publish, or preparation of the manuscript.

==============================
A phylogenetic taxonomy of species in the Cyrtodactylus peguensis group from the Ayeyarwady Basin of Myanmar is constructed based on color pattern, morphology, and molecular systematic analyses using the mitochondrial gene NADH dehydrogenase subunit 2. Newly collected samples from the type locality of C. peguensis and other localities indicate that this clade is endemic to central Myanmar and contains at least seven species, four of which are undescribed. Three species, including C. peguensis occur in the low hills of the Bago Yoma Range within the central portion of the Ayeyarwady Basin. Two of these, C. myintkyawthurai sp. nov. from the northern and central Bago Yoma and C. meersi sp. nov. which is syntopic with C. peguensis in the southern Bago Yoma are described herein. As more lowland hilly areas bordering, and within the Ayeyarwady Basin are surveyed, more new species of this group are likely to be discovered. These discoveries continue the recent surge of descriptions of new species of Cyrtodactylus that are being discovered in Myanmar.

Introduction

The Gekkota (geckos and pygopods) comprises one of the most diverse lizard lineages in the world, containing approximately 1,777 species to date (Uetz, Freed & Hošek, 2018). Within the Gekkota, Bent-toed Geckos (Cyrtodactylus Gray) are not only the most speciose genus, but the most rapidly growing in terms of the annual rate of newly described species (Uetz, Freed & Hošek, 2018). This is especially true throughout Indochina and Sundaland with the Southeast Asian nation of Myanmar being no exception. In fact, the number of new species of Cyrtodactylus discovered in Myanmar has increased from 19 to 34 (79%) in just the last year alone (see Grismer et al., 2017a, 2018a, 2018b). Notably this increase in diversity is not due to molecular analyses partitioning out species groups and adjusting the taxonomy accordingly but rather it is the result of discoveries made during recent expeditions into some of the more remote regions of the country.

One of the more enigmatic species in Myanmar, Cyrtodactylus peguensis (Boulenger, 1893), was described from two specimens (syntypes) collected by Signor L. Fea in 1887 from Palon (Hpa Lon), Bago Region in southern Myanmar. Unfortunately, one of the syntypes has been lost and apparently was never accessioned into the British Museum of Natural History (Patrick Campbell in litt. 2018). It is clear, however, that the specimen did exist and was available to Smith (1921) based on his statement that the syntypes have “two series of (6–8) large round spots on the back (referring to BMNH 1946.8.23) or (our italics) with the spots confluent transversely (referring to the specimen illustrated in Boulenger (1893)).” Following its description, C. peguensis became somewhat of an unscrutinized taxonomic repository and began to accumulate populations from throughout Indochina ranging from Vietnam to Thailand (Laidlaw, 1901; Boulenger, 1912; Annandale, 1913; Smith, 1916, 1921, 1930, 1935). C. peguensis is still reported to occur in western and southern Thailand (i.e., Taylor, 1962, 1963; Manthey & Grossmann, 1997; Chan-ard, Parr & Nabhitabhata, 2015) and some (with no voucher) have even reported it from northern Peninsular Malaysia (Ulber & Schäfer, 1989; Denzer & Manthey, 1991). Taylor (1962) recognized one of the southern Thai populations from Nakhon Si Thammarat Province as C. p. zebriacus and another from Trang Province, 70 km to the south in the same mountain range as C. p. peguensis. Subsequent authors (e.g., Manthey & Grossmann, 1997; Chan-ard et al., 1999; Das, 2010), however, implicitly considered both populations as C. p. zebriacus but did so without comment. In a recent phylogeny, Grismer et al. (2017a) noted that C. p. zebriacus is a member of a lineage they designated as the Indochinese clade and that it is only distantly related to species of the Indo-Burmese clade that contains true C. peguensis group species (i.e., C. pyinyaungensis, see discussion in Grismer et al. (2017a)), thus bringing into question the species identity of the Thai populations.

In order to construct a taxonomy of this group that is consistent with its evolutionary history, an understanding of the phylogenetic relationships of true C. peguensis from the type locality to the other populations in Myanmar and to C. p. zebriacus is paramount. Therefore, we collected two specimens from the Myin Mo Shwe Taung Pagoda, 9.5 km east of Hpa Lon village (the type locality) in the western foothills of the southern portion of the Bago Yoma mountain range as this locality constitutes the closest, most suitable, habitat for C. peguensis (Fig. 1). To this we added eight samples from Mt. Popa from the northernmost section of the Bago Yoma Range (Mandalay Region). This population resembles C. peguensis in color pattern but has been referred to as C. fea (Wood et al., 2012; Agarwal et al., 2014; Brennan et al., 2017) but later informally re-identified as C. peguensis (see discussion in Grismer et al., 2017a:3). We also included seven samples that were accessioned into the California Academy of Sciences (CAS) as C. peguensis: three from the central portion of the Bago Yoma Range (Bago Region), two from the Panlaung-Pyadalin Cave Wildlife Sanctuary (Shan State), one from Aung Ya Village (Rakhine State), and one from the Shwe Settaw Wildlife Sanctuary (Magway Region). Last, we included a sample of C. p. zebriacus from Nakhon Si Thammarat Province, Thailand and three samples of C. pyinyaungensis from Pyinyaung Village, Shan State—a species that has been hypothesized to be closely related to C. peguensis (Grismer et al., 2017a). Tissues from these samples were used to construct a molecular phylogeny based on 1,467 base pairs of the mitochondrial gene NADH dehydrogenase subunit 2 (ND2) and its flanking tRNA regions. Morphological and color pattern data taken from specimens from the type locality (including the remaining syntype), Mt. Popa, the central Bago Yoma, and C. pyinyaungensis were analyzed and the results compared to the molecular phylogeny. We also examined 16 specimens of C. p. zebriacus from three localities in southern Thailand and compared them to species of the peguensis group. The results of these analyses corroborate one another and indicate that in order to bring the taxonomy of the peguensis group in line with its evolutionary history (i.e., phylogeny), at least four new species must be described and that C. p. zebriacus should be elevated to species status being that it is not closely related to or morphologically similar to any species in the peguensis group.

Figure 1 Distribution map of the Cyrtodactylus peguensis group.

Distribution of the species of the Cyrtodactylus peguensis group in the Ayeyarwady Basin and the adjacent foothills of the Chin Hills and Shan Hills in Myanmar.

Materials and Methods

Species delimitation

The general lineage concept (GLC: De Queiroz, 2007) used herein proposes that a species constitutes a population of organisms independently evolving from other such populations owing to a lack of gene flow. By “independently,” it is meant that new mutations arising in one species cannot spread readily into another species (Barraclough, Birky & Burt, 2003; De Queiroz, 2007). Increasingly, integrative studies on the nature and origins of species are using a wider range of empirical data to delimit species boundaries (Coyne & Orr, 1998; Knowles & Carstens, 2007; Fontaneto et al., 2007; Feulner et al., 2007; Leaché et al., 2009), rather than relying solely on traditional taxonomic methods. Under the GLC herein, molecular phylogenies were used to infer species boundaries and univariate analysis of variance (ANOVA) and multivariate principal component analysis (PCA) and discriminant analysis of principal components (DAPC) of morphological data were used to describe those boundaries. These boundaries were cross-checked using a Generalized Mixed Yule Coalescent (GMYC) approach (Pons et al., 2006), thus providing an independent framework to complement the empirically based thresholds of the morphological and molecular analyses. The GMYC approach is a method for delimiting species from single-locus gene trees by detecting genetic clustering beyond the expected levels of a null hypothesis which infers that all individuals of a population form a genetically, interacting nexus. In clades where effective population sizes are relatively low and divergence times among the populations are relatively high, the single-threshold version of the model (such as that used herein) outperforms the multi-threshold version (Fujisawa & Barraclough, 2013). The GMYC relies on the prediction that independent evolution leads to the appearance of distinct genetic clusters, separated by relatively longer internal branches (Barraclough, Birky & Burt, 2003; Acinas et al., 2004). Such groups therefore, diverge into discrete units of morphological and genetic variation that are recovered with surveys of higher clades. The analysis was run on the web server at http://species.h-its.org/gmyc/.

Molecular data and analyses

The data set of Grismer et al. (2017a), which included exemplars of all the major Cyrtodactylus clades in Agarwal et al. (2014), was augmented with 17 samples purported to be C. peguensis, totaling 222 ingroup samples. Hemidactylus angulatus Hallowell, H. frenatus Duméril & Bibron, H. garnotii Duméril & Bibron, H. mabouia (Moreau de Jonnès), and H. turcicus (Linnaeus) served as outgroups following Grismer et al. (2017a). All new sequences were deposited in GenBank (Table 1). Genomic DNA was isolated from liver or skeletal muscle tissue stored in 95% ethanol using a Maxwell® RSC Tissue DNA kit on the Promega Maxwell® RSC extraction robot. ND2 was amplified using a double-stranded polymerase chain reaction (PCR) under the following conditions: 1.0 μl genomic DNA (10–30 μg), 1.0 μl light strand primer (concentration 10 μM), 1.0 μl heavy strand primer (concentration 10 μM), 1.0 μl dinucleotide pairs (1.5 μM), 2.0 μl 5× buffer (1.5 μM), MgCl 10× buffer (1.5 μM), 0.1 μl Taq polymerase (five μ/μl), and 6.4 μl ultra-pure H2O. PCR reactions were executed on Bio-Rad gradient thermocycler under the following conditions: initial denaturation at 95 °C for 2 min, followed by a second denaturation at 95 °C for 35 s, annealing at 48–52 °C for 35 s, followed by a cycle extension at 72 °C for 35 s, for 31 cycles. All PCR products were visualized using electrophoresis on a 1.0% agarose gel. Successful PCR products were sent to GENEWIZ® for PCR purification, cycle sequencing, sequencing purification, and sequencing using the same primers as in the amplification step (Table 2). Sequences were analyzed from both the 3′ and the 5′ ends separately to confirm congruence between the reads. Forward and reverse sequences were uploaded and edited in Geneious™ version v6.1.8. Following sequence editing we aligned the protein-coding region and the flanking tRNAs using the MAFTT v7.017 (Katoh et al., 2002) plugin under the default settings in Geneious™ (Kearse et al., 2012). Mesquite v3.04 (Maddison & Maddison, 2015) was used to calculate the correct amino acid reading frame and to confirm the lack of premature stop codons in the ND2 portion of the DNA fragment.

Table 1 Locality data and GenBank numbers of specimens used in this analysis.

Taxon	Catalog no.	Locality	GenBank no.	
Cyrtodactylus meersi sp. nov. holotype	LSUHC 13455	Myin Mo Shwe Taung Pagoda, Bago Region Taikkyi Township, Yangon (north) District, Myanmar (17.46272°N, 96.01624°E, elevation 162 m)	MH624104	
Cyrtodactylus myintkyawthurai sp. nov.	CAS 245200	Central Bago Yoma, between Pallan Gyi Forest Camp and summit, Bago Region Myamnar (18.92108°N, 95.814027°E, 268 m)	MH624107	
Cyrtodactylus myintkyawthurai sp. nov.	CAS 245201	Central Bago Yoma, between Pallan Gyi Forest Camp and summit, Bago Region Myamnar (18.92108°N, 95.814027°E, 268 m)	MH624117	
Cyrtodactylus myintkyawthurai sp. nov.	CAS 245202	Central Bago Yoma, between Pallan Gyi Forest Camp and summit, Bago Region Myamnar (18.92108°N, 95.814027°E, 268 m)	MH624108	
Cyrtodactylus myintkyawthurai sp. nov.	LSUHC 13500	Taung Twin Chaung camp, Mt. Popa, Kyauk-pa-taung Township, Mandalay Region, Myanmar (20.93087°N, 95.22580°E; 978 m in elevation)	MH624109	
Cyrtodactylus myintkyawthurai sp. nov.	LSUHC 13501	Taung Twin Chaung camp, Mt. Popa, Kyauk-pa-taung Township, Mandalay Region, Myanmar (20.93087°N, 95.22580°E; 978 m in elevation)	MH624116	
Cyrtodactylus myintkyawthurai sp. nov.	LSUHC 13502	Taung Twin Chaung camp, Mt. Popa, Kyauk-pa-taung Township, Mandalay Region, Myanmar (20.93087°N, 95.22580°E; 978 m in elevation)	MH624110	
Cyrtodactylus myintkyawthurai sp. nov.	LSUHC 13505	Taung Twin Chaung camp, Mt. Popa, Kyauk-pa-taung Township, Mandalay Region, Myanmar (20.93087°N, 95.22580°E; 978 m in elevation)	MH624112	
Cyrtodactylus myintkyawthurai sp. nov.	USNM 559805	Popa Mountain Park, Mandalay Region Myanmar	JX440536	
Cyrtodactylus myintkyawthurai sp. nov.	LSUHC 13806	Taung Twin Chaung camp, Mt. Popa, Kyauk-pa-taung Township, Mandalay Region, Myanmar (20.93087°N, 95.22580°E; 978 m in elevation)	MH624114	
Cyrtodactylus myintkyawthurai sp. nov. holotype	LSUHC 13807	Taung Twin Chaung camp, Mt. Popa, Kyauk-pa-taung Township, Mandalay Region, Myanmar (20.93087°N, 95.22580°E; 978 m in elevation)	MH624115	
Cyrtodactylus myintkyawthurai sp. nov.	LSUHC 13808	Taung Twin Chaung camp, Mt. Popa, Kyauk-pa-taung Township, Mandalay Region, Myanmar (20.93087°N, 95.22580°E; 978 m in elevation)	MH624113	
Cyrtodactylus myintkyawthurai sp. nov.	LSUHC 13809	Taung Twin Chaung camp, Mt. Popa, Kyauk-pa-taung Township, Mandalay Region, Myanmar (20.93087°N, 95.22580°E; 978 m in elevation)	MH624111	
Cyrtodactylus pyinyaungensis	LSUHC 13139	5.3 km north of Pyinyaung Village at the Apache Cement factory mining site, Mandalay Region, Myanmar (N20°52.191, E96°24.296; 642 m in elevation)	MH624119	
Cyrtodactylus pyinyaungensis	LSUHC 13148	5.3 km north of Pyinyaung Village at the Apache Cement factory mining site, Mandalay Region, Myanmar (N20°52.191, E96°24.296; 642 m in elevation)	MH624120	
Cyrtodactylus pyinyaungensis	LSUHC 13149	5.3 km north of Pyinyaung Village at the Apache Cement factory mining site, Mandalay Region, Myanmar (N20°52.191, E96°24.296; 642 m in elevation)	MH624121	
Cyrtodactylus peguensis	LSUHC 13454	Myin Mo Shwe Taung Pagoda, Bago Region Taikkyi Township, Yangon (north) District, Myanmar (17.46272°N, 96.01624°E, elevation 162 m)	MH624122	
Cyrtodactylus sp. 1	CAS 226142	Panlaung-Pyadalin Cave Wildlife Sanctuary, Ywa Ngan Township, Shan State, Myanmar (21.11581°N, 96.36069°E, 346 m in elevation)	MH624105	
Cyrtodactylus sp. 1	CAS 226143	Panlaung-Pyadalin Cave Wildlife Sanctuary, Ywa Ngan Township, Shan State, Myanmar (21.11581°N, 96.36069°E, 346 m in elevation)	MH624106	
Cyrtodactylus sp. 2	LSUHC 226139	Shwe Settaw Wildlife Sanctuary, Min Bu Township, Magway Region Myanmar (20.059805°N, 94.59606°E, 137 m in elevation)	MH624118	
Note:

GenBank accession numbers for the newly recorded specimens of the peguensis group used for the molecular phylogenetic analyses. Accession numbers for outgroups are in Agarwal et al. (2014) and for the other specimens of Cyrtodactylus of the Indochina clade see Grismer et al. (2017a, 2018a, 2018b).

Table 2 Primer sequences used in this study for amplification and sequencing the ND2 gene and the flanking tRNAs.

Primer name	Primer reference	Sequence	
L4437b	Macey et al. (1997)	External	5′-AAGCAGTTGGGCCCATACC-3′	
H5934	Macey et al. (1997)	External	5′-AGRGTGCCAATGTCTTTGTGRTT-3′	

Two model-based phylogenetic analyses, maximum likelihood (ML) and Bayesian Inference (BI) were employed. The ML analysis was implemented in IQ-TREE (Nguyen et al., 2015) which employs a Bayesian Information Criterion algorithm to calculate that K3P+I+G4 was the best-fit model of evolution for the tRNA, TVM+F+I+G4 the best-fit model for the first codon position, TIM3F+I+G4 for the second codon position, and GTR+F+ASC+G4 for the third codon position. Shimodaira-Hasegawa-like approximate likelihood ratio test (LR; Guindon et al., 2010) and Ultrafast Bootstrap Approximation (UFB; Hoang et al., 2017) using 1,000 bootstrap replicates were used to construct a final consensus tree. Nodes with UFB values of 95 and above and LR values of 0.80 and above were considered significantly supported. A BI analysis was implemented in MrBayes 3.2.3. on XSEDE (Ronquist et al., 2012) using Cyberinfrastructure for Phylogenetic Research (Miller, Pfeiffer & Schwartz, 2010). An approximation of the same models of evolution generated in IQ-TREE for the ML analysis were employed in the BI analysis where GTR+I+G was estimated to be the best-fit model of evolution for all codon positions and HKY+G was the model of evolution for the tRNA. Two simultaneous runs were performed with four chains, three hot and one cold. The simulation ran for 10,000,000 generations, was sampled every 1,000 generations using the Markov Chain Monte Carlo, and the first 25% of each run were discarded as burn-in. Stationarity and .p files from each run were checked in Tracer v1.6 (Rambaut et al., 2014) to ensure effective sample sizes were above 200 for all parameters. Nodes with Bayesian posterior probabilities of 0.95 and above were considered well-supported (Huelsenbeck et al., 2001; Wilcox et al., 2002). After removing outgroup taxa, MEGA7 (Kumar, Stecher & Tamura, 2016) was used to calculate uncorrected pairwise sequence divergence of the 21 ingroup samples.

Morphological data and analyses

Color pattern notes were taken from living and preserved specimens and digital images of living specimens of all possible age classes prior to preservation. Measurements were taken on the right side of the body when possible to the nearest 0.1 mm by Marta S. Grismer using dial calipers under a Leica Wild M10 stereo dissecting microscope following Grismer et al. (2017a). Measurements taken were: snout-vent length (SVL), taken from the tip of snout to the vent; tail length (TL), taken from the vent to the tip of the tail, original or regenerated; tail width (TW), taken at the base of the tail immediately posterior to the postcloacal swelling; forearm length (FL), taken on the dorsal surface from the posterior margin of the elbow while flexed 90° to the inflection of the flexed wrist; tibia length (TBL), taken on the ventral surface from the posterior surface of the knee while flexed 90° to the base of the heel; axilla to groin length (AG), taken from the posterior margin of the forelimb at its insertion point on the body to the anterior margin of the hind limb at its insertion point on the body; head length (HL), the distance from the posterior margin of the retroarticular process of the lower jaw to the tip of the snout; head width (HW), measured at the angle of the jaws; head depth (HD), the maximum height of head measured from the occiput to the throat; eye diameter (ED), the greatest horizontal diameter of the eyeball; eye to ear distance (EE), measured from the anterior edge of the ear opening to the posterior edge of the eyeball; eye to snout distance (ES), measured from anteriormost margin of the eyeball to the tip of snout; eye to nostril distance (EN), measured from the anterior margin of the eye ball to the posterior margin of the external nares; inter orbital distance (IO), measured between the anterior edges of the orbit; ear diameter (EL), the greatest vertical distance of the ear opening; and internarial distance (IN), measured between the nares across the rostrum.

Meristic characters taken were the numbers of supralabial (SL) and infralabial (IL) scales counted from the largest scale immediately below the middle of the eyeball to the rostral and mental scales, respectively; the number of paravertebral tubercles (PVT) between limb insertions counted in a straight line immediately left of the vertebral column; the number of longitudinal rows of dorsal tubercles (LRT) counted transversely across the center of the dorsum from one ventrolateral fold to the other; the number of longitudinal rows of ventral scales (VSs) counted transversely across the center of the abdomen from one ventrolateral fold to the other; and the total number of subdigital lamellae fourth toe lamellae (4TL). The total number of femoral pores (FP) in males (i.e., the sum of the number of enlarged pore-bearing femoral scales from each leg combined as a single metric; the number of precloacal pores in (PP) in males; the number of rows of post-precloacal scales (PPS) on the midline between the enlarged precloacal scales and the vent (see Grismer et al., 2017a:Fig. 4); number of body bands (BB) or large, transversely oriented, dark paravertebral spots between the nuchal loop (dark band running from eye to eye) and the hind limb insertions not including the nape or postsacral bands; the number of light caudal bands (LCB) on an original tail; and the number of dark caudal bands (DCB) on an original tail. Non-meristic morphological characters evaluated were the degree of body tuberculation—weak tuberculation referring to dorsal body tubercles that are relatively low, small, less densely packed, and weakly keeled whereas prominent tuberculation refers to tubercles that are larger, higher (raised), and prominently keeled (see Grismer et al., 2017a:Fig. 6); and the relative length to width ratio of the transversely expanded, median subcaudal scales. Color pattern characters (see Grismer et al., 2017a: Fig. 5) evaluated were the dorsal body bands bearing paired, paravertebral elements or fused medially; top of head bearing combinations of dark diffuse mottling or dark, distinct blotches overlain with a light-colored reticulating network or not; and light caudal bands encircling tail or not.

All statistical analyses were performed using the platform R v 3.2.1 (R Core Team, 2015). Given that sample sizes from the five populations tested (see below) were unequal, a Levene’s test was conducted to test for homogeneity of variances among each meristic character prior to an ANOVA. A Welsh’s ANOVA was conducted on characters with unequal variances (i.e., p-values ≥ 0.05 in the Levene’s test) to test for the presence of statistically significant mean differences (p ≤ 0.05) in the data set. Characters containing statistical differences were subjected to a Tukey honestly significance difference (HSD) test to ascertain which population pairs differed significantly from each other for those characters.

Principal component analysis (PCA) and discriminant analysis of principal components (DAPC) using the ADEGENET package in R (Jombart, Devillard & Balloux, 2010) were used to determine if the populations sampled occupied unique positions in morphospace and the degree to which their variation in morphospace coincided with the putative species boundaries inferred by the molecular phylogenetic, univariate, and GMYC analyses. PCA, implemented by the prcomp command in R, is an indiscriminate analysis plotting the overall variation among individuals (i.e., data points) while treating each independently (i.e., not coercing them into pre-defined groups). Femoral and precloacal pore counts were excluded from the PCA due to their presence in only males. For all other characters, both sexes and all size classes were used. All data used in the PCA were log transformed and scaled to their standard deviation prior to analysis in order to normalize their distribution so as to ensure characters with very large and very low values did not over-leverage the results owing to intervariable nonlinearity and to insure the data were analyzed on the basis of correlation not covariance. A vectorial representation of the multivariate data by use of a biplot analysis was implemented using the ggbiplot command in R. Here, the information for both the populations/species and the characters are overlain on the PCA plot. Groups of vectors pointing in the same direction are indicative of characters that vary in the same manner across multivariate space and the vector angles (based on either the x or y axis) are proportional to their degree of covariation. In order to determine which centroids (i.e., means) of the PCA plots differed significantly from one another, a modified version of the broken stick model (Cangelosi & Goriely, 2007) from the VEGAN package in R was implemented using the bs command. This statistical model implements a stopping rule (Jackson, 1993) in order to determine the number of interpretable principal components (PCs) (i.e., those that capture the most amount of variation in the data set with the least amount of noise) to retain for interpretation prior to PC degeneration (subsequent PCs represent negligible structure in the data and amount mostly to noise). The analysis using this model produces overlapping curves of eigenvalues and broken stick values and proposes that the number of retained PCs should have eigenvalues higher than their corresponding random broken stick components (the null model). For this analysis, first principal component (PC1) which accounts for 44% of the total variation in the data set, was the only PC retained. The data points from PC1 were subjected to a Welsh’s ANOVA and a Tukey HSD test to ascertain which pairs of species centroids differed significantly from one another.

The DAPC places the individuals of each pre-defined population into separate clusters (i.e., plots of points) bearing the smallest within-group variance that produce linear combinations of centriods having the greatest between-group variance (linear distance; Jombart, Devillard & Balloux, 2010). DAPC relies on log transformed and scaled data from the PCA as a prior step to ensure that the variables analyzed are not correlated and number fewer than the sample size. Selecting the principal components generated by the DAPC that collectively account for at least 90% of the variation in the data set are retained for the analysis (Jombart, Devillard & Balloux, 2010) as a dimension reduction mechanism to remove noise in the data set. Retaining too many variables, forces false structure to appear in the data. Retaining too few, runs the risk of missing true structure in the data set (Cangelosi & Goriely, 2007).

Museum abbreviations follow Sabaj (2016) except for LSUHC referring to the La Sierra University Herpetological Collection, La Sierra University, Riverside, CA, 92505, USA; and MS referring to Montri Sumontha, Ranong Marine Fisheries Station, Ranong 85000, Thailand. Ministry of Natural Resources and Environmental Conservation Forest Department of Myanmar provided collecting and export permits. The electronic version of this article in portable document format will represent a published work according to the International Commission on Zoological Nomenclature (ICZN), and hence the new names contained in the electronic version are effectively published under that Code from the electronic edition alone. This published work and the nomenclatural acts it contains have been registered in ZooBank, the online registration system for the ICZN. The ZooBank LSIDs (Life Science Identifiers) can be resolved and the associated information viewed through any standard web browser by appending the LSID to the prefix http://zoobank.org/. The LSID for this publication is: urn:lsid:zoobank.org:act: 9645421A-0997-487C-B26B-2030FC42D264. The online version of this work is archived and available from the following digital repositories: PeerJ, PubMed Central, and CLOCKSS.

Brigham Young University’s Institutional Animal Care and Use Committee has approved the animal use protocol for this study (protocol # 160401). Collecting permits were granted to Fauna & Flora International (FFI) by Mr. Win Naing Thaw of the Ministry of natural Resources and Environmental Conservation Forest Department. The authors worked under the umbrella of FFI.

Results

The ML and BI analyses produced trees with identical topologies and strong nodal support at nearly every node (Fig. 2). All but one of the Burmese populations included in the analyses that have been formally or informally considered conspecific with or allied to C. peguensis are recovered as a monophyletic group. C. annandalei Bauer was recovered as a closely related sister species to this monophyletic group and distantly related to all other species of the Indo-Burma clade (Fig. 2). As such, it and the other populations are referred to here as the peguensis group. The specimen (CAS 22143) not included in the peguensis group from Aung Ya Village, Rakhine State was recovered as an undescribed sister species to C. ayeyarwadyensis Bauer. Surprisingly, the two samples from the Myin Mo Shwe Taung Pagoda from 9.5 km east of the type locality of C. peguensis were not each other’s closest relatives and share a 10.3% uncorrected pairwise sequence divergence between them despite being collected within 200 m of one another (Table 3).

Figure 2 Phylogenetic relationships of the species of the Cyrtodactylus peguensis and C. oldhami groups.

Maximum likelihood consensus tree topology of Cyrtodactylus highlighting the distant phylogenetic position of the peguensis and oldhami groups. Black dots denote nodes with BPP ≥ 0.95, LR ≥ 0.80, and UFB ≥ 95. The gray dot connotes a node with BPP ≥ 0.95, LR ≤ 0.80, and UFB ≥ 95. The white dot connotes a node with BPP ≤ 0.95, LR ≤ 0.80, and UFB ≥ 95. Red arrow and red circle denote the node of the Indo-Burma clade (sensu Agarwal et al., 2014). Blue arrow and blue circle denote the node of the Indochinese clade (sensu Grismer et al. 2017a).

Table 3 Percent uncorrected pair-wise sequence divergence among individuals and species of the Cyrtodactylus peguensis group and C. zebriacus calculated from 1,467 base pairs of the mitochondrial gene ND2.

	1	2	3	4	5	6	7	8	9	10	11	12	13	14	15	16	17	18	19	20	21	22	
1. zebriacus CUMZ 2005 073054	***	***	***	***	***	***	***	***	***	***	***	***	***	***	***	***	***	***	***	***	***	***	
2. Cyrtodactylus sp.1 CAS 226139	0.303	***	***	***	***	***	***	***	***	***	***	***	***	***	***	***	***	***	***	***	***	***	
3. Cyrtodactylus sp.2 CAS 226142	0.283	0.097	***	***	***	***	***	***	***	***	***	***	***	***	***	***	***	***	***	***	***	***	
4. Cyrtodactylus sp.2 CAS 226143	0.280	0.093	0.003	***	***	***	***	***	***	***	***	***	***	***	***	***	***	***	***	***	***	***	
5. meersi sp. nov. LSUHC 13455	0.283	0.100	0.110	0.107	***	***	***	***	***	***	***	***	***	***	***	***	***	***	***	***	***	***	
6. myintkyawthurai sp. nov. Bago Yoma CAS 245200	0.293	0.117	0.097	0.093	0.113	***	***	***	***	***	***	***	***	***	***	***	***	***	***	***	***	***	
7. myintkyawthurai sp. nov. Bago Yoma CAS 245202	0.293	0.113	0.093	0.090	0.110	0.003	***	***	***	***	***	***	***	***	***	***	***	***	***	***	***	***	
8. myintkyawthurai sp. nov. Bago Yoma CAS 245201	0.303	0.123	0.103	0.100	0.117	0.013	0.010	***	***	***	***	***	***	***	***	***	***	***	***	***	***	***	
9. myintkyawthurai sp. nov. Mt. Popa LSUHC 13500	0.313	0.117	0.103	0.100	0.120	0.033	0.030	0.040	***	***	***	***	***	***	***	***	***	***	***	***	***	***	
10. myintkyawthurai sp. nov. Mt. Popa LSUHC 13502	0.313	0.117	0.103	0.100	0.120	0.033	0.030	0.040	0.000	***	***	***	***	***	***	***	***	***	***	***	***	***	
11. myintkyawthurai sp. nov. Mt. Popa LSUHC 13809	0.313	0.117	0.103	0.100	0.120	0.033	0.030	0.040	0.000	0.000	***	***	***	***	***	***	***	***	***	***	***	***	
12. myintkyawthurai sp. nov. Mt. Popa LSUHC 13806	0.313	0.117	0.103	0.100	0.120	0.033	0.030	0.040	0.000	0.000	0.000	***	***	***	***	***	***	***	***	***	***	***	
13. myintkyawthurai sp. nov. Mt. Popa LSUHC 13808	0.313	0.117	0.103	0.100	0.120	0.033	0.030	0.040	0.000	0.000	0.000	0.000	***	***	***	***	***	***	***	***	***	***	
14. myintkyawthurai sp. nov. Mt. Popa LSUHC 13807	0.313	0.117	0.103	0.100	0.120	0.033	0.030	0.040	0.000	0.000	0.000	0.000	0.000	***	***	***	***	***	***	***	***	***	
15. myintkyawthurai sp. nov. Mt. Popa LSUHC 13505	0.313	0.117	0.103	0.100	0.120	0.033	0.030	0.040	0.000	0.000	0.000	0.000	0.000	0.000	***	***	***	***	***	***	***	***	
16. myintkyawthurai sp. nov. Mt. Popa LSUHC 13501	0.313	0.117	0.103	0.100	0.120	0.033	0.030	0.040	0.000	0.000	0.000	0.000	0.000	0.000	0.000	***	***	***	***	***	***	***	
17. myintkyawthurai sp. nov. Mt. Popa USNM 559805	0.313	0.117	0.103	0.100	0.120	0.033	0.030	0.040	0.000	0.000	0.000	0.000	0.000	0.000	0.000	0.000	***	***	***	***	***	***	
18. peguensis LSUHC 13454	0.293	0.093	0.093	0.090	0.103	0.043	0.040	0.050	0.050	0.050	0.050	0.050	0.050	0.050	0.050	0.050	0.050	***	***	***	***	***	
19. pyinyaungensis LSUHC 12939	0.293	0.100	0.093	0.090	0.117	0.047	0.043	0.053	0.050	0.050	0.050	0.050	0.050	0.050	0.050	0.050	0.050	0.033	***	***	***	***	
20. pyinyaungensis LSUHC 13149	0.293	0.100	0.093	0.090	0.117	0.047	0.043	0.053	0.050	0.050	0.050	0.050	0.050	0.050	0.050	0.050	0.050	0.033	0.000	***	***	***	
21. pyinyaungensis LSUHC 13148	0.293	0.100	0.093	0.090	0.117	0.047	0.043	0.053	0.050	0.050	0.050	0.050	0.050	0.050	0.050	0.050	0.050	0.033	0.000	0.000	***	***	
22. annandalei CAS 215722	0.280	0.103	0.143	0.140	0.137	0.153	0.150	0.160	0.153	0.153	0.153	0.153	0.153	0.153	0.153	0.153	0.153	0.150	0.143	0.143	0.143	***	
Note:

*** Place holder of opposing equal values.

In comparing each specimen from the Myin Mo Shwe Taung Pagoda to the syntype of C. peguensis (BMNH 1946.8.23.10), it is clear that LSUHC 13454 is much closer morphologically to the type than is LSUHC 13455 (Table 4). The syntype and LSUHC 13454 have 17–19 longitudinal rows of dorsal tubercles (LRT) vs. 13 in LSUHC 13455, 36 or 37 vs. 32 ventral scales, 19 subdigital lamellae on the fourth toe (4TL) vs. 17, and three vs. two post-precloacal scale rows. Furthermore, the syntype and LSUHC 13454 group closely together in the PCA where 44% of the variation in the data set occurs along PC1 and loads most heavily for fourth toe lamellae and infralabial scales and 16% of the variation occurs along PC2 and loads most heavily for body bands (Table 5; Fig. 3). Additionally, the syntype and LSUHC 13454 are well-separated from LSUHC 13455 in both the PCA and DAPC analyses (Fig. 3). Therefore, we consider LSUHC 13454 to be C. peguensis and LSUHC 13455 to be a new species (described below). The analyses also recovered the Shan State specimens (CAS 226142–43) from the Pyadalin Cave region and CAS 226139 from the Shwe Settaw Wildlife Sanctuary, Magway Region to be distinct from one another and from all other peguensis group lineages and will be described elsewhere when specimens become available. The PCA and DAPC analyses recovered C. pyinyaungensis and C. peguensis as the most morphospatially distinct species in the peguensis group (Fig. 3) and the biplot analysis shows a distinct dichotomy along PC1 in character variation between the number of body bands which do not covary with any of the remaining covarying characters. All species are widely separated in the DAPC analysis.

Table 4 Meristic, mensural, and color pattern data for Cyrtodactylus peguensis and C. meersi sp. nov.

	BM	LSUHC	LSUHC	
	1946 8.23.10	13454	13455	
	Syntype		Holotype	
	peguensis	peguensis	meersi sp. nov.	
Sex	m	m	Juvenile	
Supralabials	7	7	7	
Infralabials	7	7	8	
Body tubercles low and rounded (dome-shaped)	Yes	Yes	Yes	
Body tubercles pointed and strongly keeked (triangular)	No	No	No	
Number of paravertebral tubercles	32	31	32	
Number longitudinal rows of tubercles	17	19	13	
Tubercles on at least anterior 1/3 of tail	/	No	No	
Number of ventral scales	37	36	32	
Subdigital lamellae on fourth toe	19	19	17	
Total number of femoral pores in males	19	17	12	
Number of precloacal pores in males	8	8	8	
Post-precloacal scales rows	3	3	2	
Transverse subcaudal 2–3 times wider than long	/	Yes	Yes	
Dorsal pattern with paravertebral elements	Yes	Variable	Yes	
Paravertebral elements not in contact	Yes	Variable	Yes	
Number of body bands	5	5	5	
Number of light bands on tail	/	12	11	
Number of dark bands on tail	/	13	10	
SVL	70.0	44.0	36.0	
TL	5.9	46.0	38.0	
TW	6.6	4.1	3.6	
FL	10	7.0	4.7	
TBL	12.3	7.5	6.1	
AG	30	18.5	16.2	
HL	20.2	12.3	9.7	
HW	11.5	8.0	6.7	
HD	8.9	5.4	4.0	
ED	4.2	3.0	2.6	
EE	5.7	3.2	2.9	
ES	7.7	5.2	4.0	
EN	6.2	3.3	2.8	
IO	5.1	3.0	3.4	
EL	1.9	1.1	0.9	
IN	2.5	1.7	1.4	

Table 5 Summary statistics and principal component analysis scores for the species of the peguensis group.

	PC1	PC2	PC3	PC4	PC5	PC6	PC7	PC8	
Standard deviation	1.87977	1.14550	1.04649	0.88399	0.76149	0.58707	0.45129	0.38658	
Proportion of variance	0.44171	0.16402	0.13689	0.09768	0.07248	0.04308	0.02546	0.01868	
Cumulative proportion	0.44171	0.60572	0.74261	0.84029	0.91278	0.95586	0.98132	1.00000	
Eigenvalue	3.53357	1.31218	1.09514	0.78144	0.57987	0.34466	0.20366	0.14944	
SL	−0.41562	−0.11994	0.43799	−0.04827	0.26991	−0.29239	0.58088	0.35060	
IL	−0.45874	0.08613	0.18604	0.25940	0.15835	−0.26483	−0.74375	0.17843	
PVT	−0.34827	0.40331	−0.21661	0.45143	0.04823	0.61796	0.20295	0.19983	
LRT	−0.20080	0.34158	−0.71353	−0.29496	0.03732	−0.45765	0.09269	0.16560	
VS	−0.24605	−0.56588	−0.26397	−0.43539	0.33955	0.43168	−0.17252	0.16900	
4TL	−0.49952	−0.03722	−0.03150	−0.00330	0.07709	−0.03537	0.11926	−0.85244	
PPS	−0.24112	−0.53611	−0.23620	0.34100	−0.66325	−0.13589	0.08476	0.12963	
BB	0.29492	−0.29969	−0.30170	0.57768	0.58068	−0.21672	0.09186	−0.07646	
Note:

Abbreviations are listed in the Materials and Methods.

Figure 3 PCA and DAPC analyses of the Cyrtodactylus peguensis group.

PCA biplot (A) and DAPC (B) analyses showing the morphospatial relationships of the species of the peguensis group as well as the overlap of the reciprocally monophyletic populations of C. myintkyawthurai sp. nov. from Mt. Popa (MP) and the Bago Yoma (BY).

As predicted based on morphology and color pattern (Grismer et al., 2017a), C. pyinyaungensis is the sister species of C. peguensis. Together, these sister species are most closely related to sister populations from the central Bago Yoma, Bago Region and Mt. Popa from the northern Bago Yoma, Mandalay Region These reciprocally monophyletic sister populations overlap in morphospace (Fig. 3) and are not morphologically distinguishable from one another (Table 6) nor do the centroids of their plots differ significantly (p = 0.60). They share an uncorrected pairwise sequence divergence of 3.0–4.0% despite the samples being separated by approximately 230 km (Table 3). This is in stark contrast to C. peguensis (LSUHC 13454) and LSUHC 13455 from the Myin Mo Shwe Taung Pagoda which share a 10.3% sequence divergence yet were found only 200 m apart. The central Bago Yoma and Mt. Popa populations are morphologically distinct from all other species in the peguensis group (Tables 7 and 8) and differ from them by an uncorrected pairwise sequence divergence of 4.0–16.0% (Table 3). Despite the fact that Mt. Popa is an isolated volcano delimiting the disjunct northern limit of the Bago Yoma, the intervening lowlands contain scattered, low-lying hills and suitable habitat that likely provide opportunities for gene flow or at least very recent gene flow. As such, we consider these populations conspecific and are described below as a new species.

Table 6 Meristic, mensural, and color pattern data for Cyrtodactylus myintkyawthurai sp. nov.

	LSUHC	LSUHC	LSUHC	LSUHC	LSUHC	LSUHC	LSUHC	LSUHC	
	13808	13502	13501	13500	13505	13807	13809	13806	
	Holotype	Paratype	Paratype	Paratype	Paratype	Paratype	Paratype	Paratype	
	myintkyawthurai sp. nov.	myintkyawthurai sp. nov.	myintkyawthurai sp. nov.	myintkyawthurai sp. nov.	myintkyawthurai sp. nov.	myintkyawthurai sp. nov.	myintkyawthurai sp. nov.	myintkyawthurai sp. nov.	
	Mt. Popa	Mt. Popa	Mt. Popa	Mt. Popa	Mt. Popa	Mt. Popa	Mt. Popa	Mt. Popa	
Sex	m	f	m	m	f	m	f	m	
Supralabials	6	6	6	7	7	7	7	7	
Infralabials	6	7	7	7	7	7	7	7	
Body tubercles low and rounded (dome-shaped)	No	No	No	No	No	No	No	No	
Body tubercles pointed and keeked (triangular)	Yes	Yes	Yes	Yes	Yes	Yes	Yes	Yes	
Number of paravertebral tubercles	31	33	33	31	32	33	31	30	
Number longitudinal rows of tubercles	17	21	23	21	19	18	17	18	
Number of ventral scales	35	34	35	32	32	35	32	33	
Subdigital lamellae on fourth toe	17	18	17	17	19	18	19	18	
Enlarged femoral scales and precloacal scales continuous	/	/	/	/	/	/	/	/	
Total number of femoral pores	12	Dimples	16	20	0 (juv)	14	8	12	
Number of precloacal pores in males	7	5	8	8	0 (juv)	7	8	8	
Post-precloacal scales rows	2	2	2	2	2	2	2	2	
Transverse subcaudals 2–3 times wider than long	No	/	No	No	No	No	No	No	
Dorsal pattern with paravertebral elements	Yes	Variable	No	Yes	Yes	Yes	Yes	Yes	
Paravertebral elements not in contact	No	No	No	No	No	No	No	No	
Number of body bands	4	6	5	4	4	6	5	4	
Number of light bands on tail	11	/	/	/	9	12	/	/	
Number of dark bands on tail	10	/	/	/	10	11	/	/	
SVL	57.6	75.1	63	58.8	41.7	57	53.3	51.2	
TL	64	r	r	r		68	r	r	
TW	5.9	6.7	7.3	6.4	3.5	5	5.1	4.7	
FL	9	9.9	8.4	8.2	7	8.3	7.3	7.9	
TBL	8.8	12.3	10.4	10.3	7.4	10.1	9	8.8	
AG	27.2	35.8	27.3	24.2	18.1	27.7	23.7	23.3	
HL	16.1	18.4	16.6	17.1	12.4	15.5	14.8	13.5	
HW	11.3	14.1	10.7	11.2	7.7	10.8	10.1	9.6	
HD	7	8.8	7.4	6.8	4.9	6.1	6.7	6.5	
ED	4.3	4.5	3.6	3.3	2.7	3.3	4.1	4	
EE	4.5	5.8	4.2	4.6	3.3	4.3	4	4	
ES	6.4	8.2	6.8	6.3	5	6.3	6.7	5.9	
EN	5.6	5.6	4.5	4.5	3.2	4.7	5.9	4.7	
IO	4.7	6.7	5.2	4.7	3.6	5	4.8	3.9	
EL	2.1	2.1	1.7	2.1	1.4	1.8	2.1	1.9	
IN	2.1	2.7	2.4	2.2	2	2.2	2.3	1.8	
	CAS	CAS	CAS	CAS	CAS	CAS	
	245200	245202	222147	245203	222128	245201	
	Paratype	Paratype	Paratype	Paratype	Paratype	Paratype	
	myintkyawthurai sp. nov.	myintkyawthurai sp. nov.	myintkyawthurai sp. nov.	myintkyawthurai sp. nov.	myintkyawthurai sp. nov.	myintkyawthurai sp. nov.	
	Bago Yoma	Bago Yoma	Bago Yoma	Bago Yoma	Bago Yoma	Bago Yoma	
Sex	f	f	f	f	m	m	
Supralabials	7	7	7	7	7	7	
Infralabials	6	6	6	6	6	6	
Body tubercles low and rounded (dome-shaped)	No	No	No	No	No	No	
Body tubercles pointed and keeked (triangular)	Yes	Yes	Yes	Yes	Yes	Yes	
Number of paravertebral tubercles	28	29	32	30	32	31	
Number longitudinal rows of tubercles	17	19	17	17	18	18	
Number of ventral scales	36	35	35	36	34	35	
Subdigital lamellae on fourth toe	17	18	17	17	17	18	
Enlarged femoral scales and precloacal scales continuous	No	No	No	No	No	No	
Total number of femoral pores	4	1	0	1	16	13	
Number of precloacal pores in males	9	5	8	7	9	9	
Post-precloacal scales rows	2	2	2	2	2	2	
Transverse subcaudals 2–3 times wider than long	/	Yes	Yes	Yes	/	Yes	
Dorsal pattern with paravertebral elements	Yes	Yes	Yes	Yes	Yes	Yes	
Paravertebral elements not in contact	No	No	No	Yes	Variable	No	
Number of body bands	5	5	5	5	∼5	5	
Number of light bands on tail	/	/	11	11	/	11	
Number of dark bands on tail	/	/	11	12	/	11	
SVL	70	69.6	68.3	55.3	61.3	47.6	
TL	r	r	76.1	58.5	b	50.7	
TW	6	5.4	6.3	4.8	6.5	4.6	
FL	9.8	10	9.4	8.6	9.1	7.6	
TBL	11.6	10.5	11.6	9.5	10.6	8.8	
AG	30.6	34.4	34.8	22.5	26.8	23.4	
HL	18.8	19.1	18.3	14.7	16.8	13.6	
HW	11.6	11.2	11.9	9.7	10.5	8.2	
HD	7.5	7.7	9.4	5.8	6.8	5.4	
ED	4.5	4.2	4.7	3.9	4.4	4.1	
EE	4.3	4.8	5.3	3.9	4.4	3.3	
ES	6.8	6.8	6.9	6.2	6.6	4.7	
EN	4.6	5.4	5.4	4.5	5.3	3.4	
IO	5	4.5	4.5	3.8	3.9	3.1	
EL	2	2.2	1.6	1.9	2.4	1.5	
IN	2.4	2.3	2.2	1.7	2.4	1.8	
Note:

All measurements are in mm.

r = regenerated; / = data unobtainable.

Table 7 Summary statistics and diagnostic characters of the species from the peguensis species groups.

	peguensis	meersi sp. nov.	myintkywathurai sp. nov.	pyinyaungensis	annandalei*	
Supralabial scales (SL)						
Mean	7	7	6.9	8.0	/	
SD	0	0	±0.36	0	/	
Range	7	7	6 or 7	8	7 or 8	
N	2	1	14	5	3	
Infralabial scales (IL)						
Mean	7	8	6.5	6	9	
SD	0	0	±0.52	0	0	
Range	7	8	6 or 7	6	9	
N	2	1	14	5	3	
Paravertebral tubercles (PVT)						
Mean	31.5	32	31.1	28.2	/	
SD	±0.7	0	±1.51	2.49	/	
Range	31 or 32	32	28–33	25–30	/	
N	2	1	14	5	/	
Longitudinal rows of dorsal tubercles (LRT)						
Mean	18.0	13	18.6	16.4	/	
SD	±1.41	0	±1.87	1.14	/	
Range	17–19	13	17–23	15–18	16–18	
N	2	1	14	5	3	
Ventral scales (VS)						
Mean	36.5	32	32.4	31.8	/	
SD	±0.71	0	±01.42	2.49	/	
Range	36 or 37	32	32–36	30–36	43	
N	2	1	14	5	3	
Fourth toe lamellae (4TL)						
Mean	19.0	17.0	17.6	15.0	/	
SD	0	0	±0.74	0	/	
Range	19	17	17–19	15	10	
N	2	1	14	5	3	
Femoral pores in males (FP)						
Mean	18.0	12	14.9	17.5	/	
SD	±1.00	0	±2.97	±1.00	/	
Range	17–19	12	12–20	17 or 18	20–22	
N	2	1	7	2	3	
Precloacal pores in males (PP)						
Mean	8	8	8	8	/	
SD	0	0	±0.82	0	/	
Range	8	8	7–9	8	11 or 12	
N	2	1	2	2	3	
Enlarged post-precloacal scales (PPS)						
Mean	3	2	2	2	/	
SD	0	0	0	0	/	
Range	3	2	2	2	2	
N	2	1	14	5	3	
Body bands (BB)						
Mean	5	5	4.9	5.8	/	
SD	0	0	±0.66	0.45	/	
Range	5	5	4–6	5 or 6	4	
N	2	1	14	5	3	
Light caudal bands (LCB)						
Mean	12.5	11	10.8	10.8	/	
SD	±0.50	0	±0.98	1.71	/	
Range	12 or 13*	11	9–12	9–13	11	
N	2	1	6	4	1	
Dark caudal bands (DCB)						
Mean	13	10	10.8	11.0	/	
SD	0	0	±0.75	1.63	/	
Range	13	10	10–12	9–13	12	
N	2	1	6	4	1	
Morphology						
Body tubercles domed to weakly conical and weakly keeled	Yes	Yes	No	No	Yes	
Body tubercles raised, moderately to strongly keeled	No	No	Yes	Yes	No	
Color pattern						
Top of head patternless or blotched	Blotched	Blotched	Blotched	Blotched	Patternless	
Paravertebral elements of dorsal bands confluent	Variable	Yes	Variable	Yes	Yes	
Maximum SVL (mm)	70	36.0	75.1	71.7	55	
Notes:

SD, standard deviation; N, sample size.

* Data come from Bauer (2003).

Table 8 Pairwise comparison of characters with statistically significant mean differences among species of the peguensis group.

	meersi sp. nov.	myintkyawthurai sp. nov.	peguensis	pyinyaungensis	
meersi sp. nov.	***	***	***	***	
myintkyawthurai sp. nov.	IL	***	***	***	
peguensis	LRT	PPS, 4TL	***	***	
pyinyaungensis	IL, 4TL	BB, IL, PV, 4TL	IL, LRT, PPS, 4TL, VS	***	
Notes:

The raw data for Cyrtodactylus annandaeli were unavailable.

*** Place holder of opposing equal values.

The GMYC species delimitation independently recovered the same species inferred from the above analyses with a significant likelihood ratio of 40.59428 (p = 1.52131e-09). Sukumaran & Knowles (2017) demonstrated that species delimitation methods generally overestimate species diversity by recovering clades not species and that additional criteria such as morphology should be used in conjunction with these analyses. Fujisawa & Barraclough (2013) also note that the GMYC approach should be used in conjunction with additional independent data. We agree with these recommendations and believe the GMYC recovered noteworthy interpopulational genetic structure among individuals collected just meters apart that corroborates species boundaries characterized by the morphological data. Therefore, we base our species delimitations on a more integrative approach.

As previously hypothesized on the basis of molecular evidence (Grismer et al., 2017a, 2018a, 2018b), the phylogenetic analyses indicate that C. peguensis zebraicus of southern Thailand is not closely related to C. peguensis but nested within the oldhami group (sensu Connette et al., 2017) of the Indochinese clade and not with species of the peguensis group of the Indo-Burmese clade (Fig. 2). It is shares a sequence divergence of 28.0–31.3% among species of the peguensis group (Table 3). Therefore, we remove C. p. zebriacus from the synonymy of C. peguensis and elevate it to the full species, C. zebriacus Taylor, 1962. Based on the examination of eight specimens of C. zebriacus from Ko Samui Island, Surat Thani Province (CAS 23568–75), six specimens from Trang Province (FMNH 176851–54, 176857, 177328), and two specimens from Nakhon Si Thammarat, Province (FMNH 178286 and 215984), C. zebriacus differs from all members of the peguensis group by having larger and more strongly keeled body tubercles; no femoral pores; a dorsal pattern that is generally banded or has a more reticulated pattern as opposed to being composed of large, dark, paravertebral spots or hour glass-shaped bands; the top of the head bears a diffuse, reticulated pattern as opposed to having large, dark, well-defined spots; having black and white caudal bands of nearly equal width as opposed to having dark-brown and yellow caudal bands of unequal width; and the light-colored (i.e., white) caudal bands encircle the tail as opposed to being incomplete in the subcaudal region (Fig. 4).

Figure 4 Cyrtodactylus zebriacus and C. pyinyaungensis.

(A) Uncollected specimen of Cyrtodactylus zebriacus (LSUDPC 10080) from Kanchandt, Surat Thani Province, Thailand. (B) Cyrtodactylus pyinyaungensis (LSUHC 13150) from Pyinyaung Village, Mandalay Division, Myanmar. Photos by L. Lee Grismer.

Systematics and Taxonomy

Below we define and diagnose the C. peguensis group and describe two of the new species recovered in the phylogenetic and morphological analyses. We also re-describe C. peguensis based on one of the syntypes (BM 1946.8.23.10) and the newly collected specimen (LSUHC 13454) from 9.5 km east of the type locality. We do this in order to correct errors and omissions in the original description (Boulenger, 1893) and subsequent descriptions of the syntypes (Smith, 1921) as well as to present new characters used herein to diagnose different species in the peguensis group.

Cyrtodactylus peguensis group

Definition and diagnosis. The C. peguensis group ranges throughout the low hills of the Ayeyarwady Basin and its low hilly margins from the Alaungdaw Kathapa National Park, Magwe Region in the northeastern foothills of the Chin Hills, eastward to the Panlaung-Pyadalin Cave Wildlife Sanctuary, Shan State in the northwestern foothills of the Shan Hills, and southward through the Bago Yoma Range to the Myin Mo Shwe Taung Pagoda, Bago Region in the southern Bago Yoma Range (Fig. 1). This clade is composed of four nominal species C. peguensis (Boulenger), C. annandalei Bauer, two new species from the Bago Yoma range (see below), C. pyinyaungensis Grismer et al., two undescribed species Cyrtodactylus sp. 1 and Cyrtodactylus sp. 2 and is defined by the following range of characters: seven or eight supralabials, 28–33 paravertebral scales, 13–23 longitudinal rows of dorsal tubercles, 32–43 ventral scales, 10–19 fourth toe lamellae, 12–22 femoral pores in males, 7–12 precloacal pores in males, two or three post-precloacal pores, 4–6 dark transverse body bands between limb insertions usually bearing paravertebral elements, 9–13 LCB and DCB, dorsal body tubercles domed and weakly keeled and conical to raised and moderately to strongly keeled, top of the head blotched or patternless never bearing a reticulated pattern, and a maximum SVL of at least 55–75.1 mm (Table 7).

Cyrtodactylus peguensis (Boulenger, 1893)

Pegu Bent-toed Gecko

(Fig. 5)

Figure 5 Type specimens and additional specimen of Cyrtodactylus peguensis.

Cyrtodactylus peguensis. (A) Boulenger’s (1893) illustration of the lost syntype from the type locality of Hpa Lon, Bago Region Myanmar. (B) Syntype BM 946.8.23.10 from the type locality. (C) LSUHC 13454 from the Myin Mo Shwe Taung Pagoda, 9.5 km east of Hpa Lon, Bago Region Myanmar. Photos by L. Lee Grismer.

Gymnodactylus peguensis Smith, 1921:29; 1935:52 in part. Wermuth, 1965:63 in part.

Cyrtodactyuls peguensis Taylor, 1963:728 in part; Denzer & Manthey, 1991:314 in part; Cox, Van Dijk & Nabhitabhata, 1998:87 in part; Pianka & Vitt, 2003:175 in part; Manthey & Grossmann, 1997:225 in part; Das, 2010:213 in part; Grismer et al., 2017a:91 in part; Brennan et al., 2017:3, in part.

Cyrtodactylus (Cyrtodactylys) peguensis Rösler, 2000:66 in part.

Other synonymies exist in several popular herpetocultural outlets but are not listed here.

Syntype. Adult male BM 1946.8.23.10 collected in 1887 by Signor L. Fea from “Palon” (Hpa Lon), “Pegu” (Bago Region), Taikkyi Township, Yangon (north) District, Myanmar. Hpa-Lon is a small village in the Ayeyarwady Basin 9.5 km west of the western foothills of the southern portion of the Bago Yoma Range where Fea reported making zoological collections (Fea, 1897). Being that foothills are the closest suitable habitat for C. peguensis east of Pa-Lon, we restrict the type locality to the Myin Mo Swhe Taung Pagoda, Bago Region, Taikkyi Township, Yangon (north) District, Myanmar (17.46272°N, 96.01624°E, elevation 162 m) situated within these foothills where we collected an additional specimen (LSUHC 13454). The other syntype could not be located.

Additional specimen. Subadult male (LSUHC 13454) collected by Myint Kwaw Thura on May 18, 2017 along a dirt road just below the Myin Mo Swhe Taung Pagoda.

Diagnosis. Cyrtodactylus peguensis differs from other species of the peguensis group by having the unique combination of seven supralabial and infralabial scales; 31 or 32 paravertebral tubercles; 17–19 longitudinal rows of dorsal tubercles; 36 or 37 ventral scales; 19 subdigital lamellae on the fourth toe; 17–19 femoral pores in males; eight precloacal pores in males; three rows of post-precloacal scales; and domed to weakly conical and weakly keeled body tubercles; and a maximum SVL of 70 mm (Table 7).

Description based on BM 1946.8.23.10 and LSUHC 13454. Maximum SVL 70.0 mm; head moderate in length (HL/SVL 0.28–0.29), wide (HW/HL 0.57–0.65), somewhat flattened (HD/HL 0.44), distinct from neck, triangular in dorsal profile; lores inflated, prefrontal region concave, canthus rostralis rounded; snout elongate (ES/HL 0.38–0.42), rounded in dorsal profile; eye large (ED/HL 0.21–0.24); ear opening triangular, moderate in size (EL/HL 0.09); eye to ear distance greater than diameter of eye; rostral rectangular, partially divided dorsally, bordered posteriorly by large left and right supranasals contacting on midline or separated by small internasal, laterally by first supralabials; external nares bordered anteriorly by rostral, dorsally by large supranasal, posteriorly by two small postnasals, ventrally by first supralabial; seven (R,L) rectangular supralabials extending to below midpoint of eye; seven (R,L) infralabials tapering smoothly to below posterior margin of orbit; scales of rostrum and lores flat, larger than granular scales on top of head and occiput; scales on top of head and occiput intermixed with slightly enlarged tubercles; dorsal supraciliaries not elongate or keeled; mental triangular, bordered laterally by first infralabials and posteriorly by large, left and right trapezoidal postmentals that contact medially for 40–60% of their length posterior to mental; one row of slightly enlarged chinshields extending posteriorly to fourth infralabial; and gular and throat scales small, granular, grading posteriorly into larger, flatter, smooth, subimbricate to imbricate, pectoral, and ventral scales.

Body relatively short (AG/SVL 0.42–0.43) with weak ventrolateral folds; dorsal scales small, interspersed with small, domed to weakly conical, semi-regularly arranged weakly keeled tubercles; tubercles extend from occiput onto base of tail but no farther; tubercles on occiput and nape small, those on posterior portion of body larger and keeled; approximately 17–19 longitudinal rows of dorsal tubercles; 31 or 32 paravertebral tubercles; approximately 36 or 37 flat, imbricate, ventral scales larger than dorsal scales composing 36 or 37 rows; eight pore-bearing precloacal scales; three rows of large post-precloacal scales; and no deep precloacal groove or depression.

Forelimbs moderate in stature, relatively short (FL/SVL 0.14–0.16); flat scales of forearm larger than those on body, not interspersed with tubercles; palmar scales flat to rounded and slightly raised; digits well-developed, relatively short, inflected at basal, interphalangeal joints; digits slightly narrower distal to inflections; claws well-developed, sheathed by a dorsal and ventral scale; hind limbs more robust than forelimbs, moderate in length (TBL/SVL 0.17–0.18), covered dorsally by granular scales interspersed with slightly larger, weakly keeled tubercles and anteriorly by large, flat, imbricate scales; ventral scales of femora flat, imbricate, larger than dorsals, lacking a row of enlarged femoral scales; 8 or 9 (R) and 9 or 10 (L) pore-bearing femoral scales; small postfemoral scales form an abrupt union with large, flat ventral scale of posteroventral margin of thigh; subtibial scales flat, imbricate; plantar scales granular, slightly raised; digits relatively short, well-developed, inflected at basal, interphalangeal joints; 19 subdigital lamellae on fourth toe; and claws well-developed, base of claw sheathed by a dorsal and ventral scale; two or three enlarged postcloacal tubercles at base of tail; and postcloacal scales flat.

Regenerated tail (BM 1946.8.23.10), approximately 70.0 mm in length, 5.9 mm in width at base, tapering to a point; dorsal scales flat, imbricate, becoming larger posteriorly; subcaudal scales slightly larger than dorsal scales; two enlarged postcloacal tubercles at base of tail; and postcloacal scales flat. Original tail (LSUHC 13454) moderate in proportions, 46.0 mm in length, 3.9 mm in width at base, tapering to a point; dorsal scales of base of tail granular rapidly becoming flatter posteriorly; and median row of transversely enlarged subcaudal scales twice as wide as long not extending onto lateral margin of tail.

Coloration of LSUHC 13455 life (Fig. 5). Dorsal ground color of head body, and limbs light-brown; dorsal ground color of tail yellow; top of head bearing large, dark-brown, round blotches edged in yellow; dark-brown, wide, nuchal loop extending from posterior margin of one eye, across occiput, to posterior margin of other eye; nape bearing a large, dark-brown band edged in yellow with distinct paravertebral sections; five dark-brown body bands edged in yellow with distinct paravertebral sections between limb insertions; one dark-brown post-sacral band edged in yellow bearing a longitudinal yellow stripe; 13 dark-brown caudal bands wider than 12 yellow, hourglass-shaped, caudal bands with darkened centers; dorsal portion of forelimbs mottled with yellow; dorsal portion of hind limbs bearing large, elliptical, dark-brown blotches edges in yellow; flanks with a series of 9–11 dark-brown, round to irregularly shaped blotches edged in yellow. All ventral surfaces generally beige, immaculate except for ventral surfaces of forelimbs, forelegs, hands, feet, and posterior section of tail bearing dark pigment.

Distribution. Cyrtodactylus peguensis is known only from the type locality of Myin Mo Shwe Taung Pagoda, 9.5 km east of the village of Hpa Lon, Bago Region, Taikkyi Township, Yangon (north) District Myanmar (Fig. 1).

Natural History. No natural history data accompanied the description of the syntypes. LSUHC 13455 was collected in a region composed of low foothills and highly disturbed forest (Fig. 6). The specimen was collected at 2,100 h as it was crawling down an earthen bank covered with small rocks (<1 m in diameter) along the side of steep dirt road at the base of the Myin Mo Shwe Taung Pagoda.

Figure 6 Type locality of Cyrtodactylus peguensis.

Habitat and type locality of Cyrtodactylus peguensis on the road just below the Myin Mo Shwe Taung Pagoda, Bago Region, Myanmar. Photo by L. Lee Grismer.

Comparisons. We did not have access to the specimens from the Panlaung-Pyadalin Cave Wildlife Sanctuary, Shan State (CAS 226142–43) or the Shwe Settaw Wildlife Sanctuary, Magway Region (CAS 226139) that are illustrated in the phylogeny, and thus could not compare them to C. peguensis or the new species described herein. It is clear however, that based on their phylogenetic relationships they are distinct from all the species discussed in this report. C. peguensis is differentiated from C. annandalei by the top of the head being blotched as opposed to being patternless. C. peguensis is most closely related to C. pyinyaungensis (Fig. 2) but differs from it by having significantly higher mean values of infralabial scales (IL; 7.0 vs. 6.0), longitudinal rows of dorsal tubercles (LRT; 18.0 vs. 16.4), post-precloacal scales (PPS; 3.0 vs. 2.0), fourth toe lamellae (4TL; 19.0 vs. 17.0), and ventral scales (VS; 36.5, vs. 31.8) (Tables 7 and 8). C. peguensis further differs from the new species (see below) from Mt. Popa and the Bago Yoma by having significantly higher mean values of post-precloacal scales (PPS 3.0 vs. 2.0), and fourth toe lamellae (4TL; 19.0 vs. 17.6) and domed to weakly conical and weakly keeled body tubercles vs. raised and moderately keeled body tubercles (Tables 6 and 7). From the new species (see below) known only from the Myin Mo Swhe Taung Pagoda, it differs by having a significantly higher mean number of longitudinal rows of dorsal body tubercles (LRT; 18.0 vs. 13.0) and significantly more post-precloacal scales (PPS; 3.0 vs. 2.0) (Tables 7 and 8). C. peguensis is widely separated from C. pyinaungensis and C. meersi sp. nov. in the PCA and DAPC analyses and shares a 3.3 and 10.3% uncorrected pairwise sequence divergence, respectively, from them. It shares a 4.0–5.0% sequence divergence with C. myintkyawthurai sp. nov. (Table 3).

Remarks. Boulenger (1893) listed C. peguensis as having nine supralabials and seven or eight infralabials and Smith (1921) reports 9–11 supralabials and 7–9 infralabilas whereas we report seven supralabials and infralabials in the syntype. Boulenger’s and Smith’s totals were a result of counting all the granular scales along the labial margins posterior to the enlarged labial scales which was not done here. Boulenger (1893) reports the syntypes as having 43–45 ventral scales across the belly whereas we report 37 scales in the syntype between the ventrolateral folds. Boulenger (1893) did not indicate how his counts were made. Boulenger (1893) states that femoral pores were absent in the syntypes, however, we counted nine femoral pores on the right leg and 10 on the left. Femoral pores in species of the peguensis group are difficult to see because they are small and not imbedded within enlarged femoral scales as in other Cyrtodactylus. Additionally, the pores on the pore-bearing scales generally lie hidden beneath the posterior edges of the scales of the scale row immediately anterior to the pore-bearing row. Boulenger (1893) described the dorsal pattern as consisting of two series of large dorsal spots. This is true for the syntype BMNH 1946.8.23.10 however based on his illustration of the lost syntype, the spots are confluent on the midline as they are in LSUHC 13454 (Fig. 5). This description was made even more confusing by Smith (1921: 428) who accurately noted the difference between the syntypes but provides an illustration of a specimen of C. zebriacus from southern Thailand bearing a very different color pattern which he referred to as the “Forma typica.”

Cyrtodactylus meersi sp. nov.

Bago Yoma Bent-toed Gecko

(Fig. 7)

urn:lsid:zoobank.org:act:172923F3-0422-4247-A5D0-38E30FE1B1C5

Figure 7 Holotype of Cyrtodactylus meersi.

Holotype of Cyrtodactylus meersi sp. nov. (LSUHC 13455) from the type locality of the Myin Mo Shwe Taung Pagoda, Bago Division, Myanmar. Photo by L. Lee Grismer.

Holotype. Juvenile male LSUHC 13455 collected on May 18, 2017 at 2,000 h by Mark. W. Herr from Myin Mo Swhe Taung Pagoda, Bago Region, Taikkyi Township, Yangon (north) District, Myanmar (17.46272°N, 96.01624°E, elevation 162 m).

Diagnosis. Cyrtodactylus meersi sp. nov. differs from other species of the peguensis group by having the unique combination of seven supralabials and eight infralabials; 32 paravertebral tubercles; 13 longitudinal rows of body tubercles; 32 ventral scales; 17 subdigital lamellae on the fourth toe; 12 femoral pores; eight precloacal pores; two rows of post-precloacal scales; and domed to weakly conical and weakly keeled body tubercles (Table 7). We note, however, that this diagnosis is not robust due to having only a sample size of one juvenile and will be subject to adjustment if additional specimens are ever collected and analyzed. Nonetheless, the placement of this individual near the base of the phylogeny (Fig. 2) and it having an uncorrected percent sequence divergence of 10.0–13.7% from all other species in the phylogeny (Table 3) is strong evidence of its species status.

Description of holotype. Juvenile, SVL 36.0 mm; head moderate in length (HL/SVL 0.27), wide (HW/HL 0.67), somewhat flattened (HD/HL 0.41), distinct from neck, triangular in dorsal profile; lores inflated, prefrontal region concave, canthus rostralis rounded; snout elongate (ES/HL 0.42), rounded in dorsal profile; eye large (ED/HL 0.27); ear opening elliptical to triangular, moderate in size (EL/HL 0.09); eye to ear distance greater than diameter of eye; rostral rectangular, partially divided dorsally, bordered posteriorly by large left and right supranasals separated by small internasal, laterally by first supralabials; external nares bordered anteriorly by rostral, dorsally by large supranasal, posteriorly by two small postnasals, ventrally by first supralabial; seven (R,L) rectangular supralabials extending to below midpoint of eye; eight (R,L) infralabials tapering smoothly to below posterior margin of orbit; scales of rostrum and lores flat, larger than granular scales on top of head and occiput; scales on top of head and occiput intermixed with slightly enlarged tubercles; dorsal supraciliaries not elongate or keeled; mental triangular, bordered laterally by first infralabials and posteriorly by large, left and right trapezoidal postmentals that contact medially for 60% of their length posterior to mental; one row of slightly enlarged chinshields extending posteriorly to fourth infralabial; and gular and throat scales small, granular, grading posteriorly into larger, flatter, smooth, subimbricate to imbricate, pectoral, and ventral scales.

Body relatively short (AG/SVL 0.45) with weak ventrolateral folds; dorsal scales small, interspersed with small, domed to weakly conical, semi-regularly arranged weakly keeled tubercles; tubercles extend from occiput onto base of tail but no farther; tubercles on occiput and nape small, those on posterior portion of body larger and keeled; approximately 13 longitudinal rows of dorsal tubercles; 32 paravertebral tubercles; approximately 32 flat, imbricate, ventral scales larger than dorsal scales; eight pore-bearing precloacal scales; two rows of large post-precloacal scales; and no deep precloacal groove or depression.

Forelimbs moderate in stature, relatively short (FL/SVL 0.13); flat scales of forearm larger than those on body, not interspersed with tubercles; palmar scales rounded, slightly raised; digits well-developed, relatively short, inflected at basal, interphalangeal joints; digits slightly narrower distal to inflections; claws well-developed, sheathed by a dorsal and ventral scale; hind limbs more robust than forelimbs, moderate in length (TBL/SVL 0.17), covered dorsally by granular scales interspersed with slightly larger, weakly keeled tubercles and anteriorly by large, flat, imbricate scales; ventral scale of femora flat, imbricate, larger than dorsals, lacking a row of enlarged femoral scales; six (R,L) weakly developed, pore-bearing femoral scales; small postfemoral scales form an abrupt union with large, flat ventral scales of posteroventral margin of thigh; subtibial scales flat, imbricate; plantar scales flat but slightly raised; digits relatively short, well-developed, inflected at basal, interphalangeal joints; 17 subdigital lamellae on fourth toe; and claws well-developed, base of claw sheathed by a dorsal and ventral scale; two enlarged postcloacal tubercles at base of tail; and postcloacal scales flat.

Original tail moderate in proportions, 38.0 mm in length, 3.6 mm in width at base, tapering to a point; dorsal scales of base of tail granular rapidly becoming flatter posteriorly; and median row of transversely enlarged subcaudal scales twice as wide as long, not extending onto lateral margin of tail.

Coloration in life (Fig. 7). Dorsal ground color of head body, limbs, and tail straw-colored; top of head bearing small, dark-brown spots, those in center of head well-defined and edged in yellow; dark-brown, wide, nuchal loop extending from posterior margin of one eye, across occiput, to posterior margin of other eye; three large, dark-brown spots edged in yellow on nape; five dorsal bands between limb insertions; anteriormost band composed of the three transversely arranged, dark-brown blotches edged in yellow; remaining bands composed of paired, separate, paravertebral, dark-brown blotches edged in yellow; postsacral band composed of three, dark-brown blotches edge in yellow; dorsal surface of limbs mottled with yellow spots and larger, poorly defined, dark blotches; 13 dark bands on tail, anteriormost bifurcated medially; 13 light-colored, yellowish bands on tail; tail-tip white; flanks bearing 8–10 small, diffuse, dark spots. All ventral surfaces generally beige, immaculate except for ventral surfaces of forelimbs, forelegs, hands, feet, and posterior section of tail bearing dark pigment. Based on ontegenetic changes in color pattern observed in C. pyinyaungensis, it is likely that with an increase in SVL, blotches on the flanks and the top of the head would become slightly larger and more defined.

Distribution. Cyrtodactylus meersi sp. nov. is known only from the type locality of Myin Mo Shwe Taung Pagoda, 9.5 km east of the village of Hpa Lon, Bago Region, Taikkyi Township, Yangon (north) District Myanmar (Fig. 1).

Etymology. The specific epithet, meersi, is named in honor of Mr. John Meers whose generous private donations to Fauna & Flora International’s in the name of karst conservation have resulted in the continuation of karst biology research in Indochina.

Natural History. The holotype was collected in a region composed of low foothills and highly disturbed forest (Fig. 6). The specimen was encountered at 2,000 h as it was sitting in the middle of an ant trail, presumably preying on the ants. The fact that the specimen is a juvenile suggests the reproductive season is prior to May.

Comparisons. Cyrtodactylus meersi sp. nov. is differentiated from C. annandalei by the top of the head being blotched as opposed to being patternless. Differences between C. meersi sp. nov. and C. peguensis are reported in the comparisons section of the latter. C. meersi sp. nov. differs from C. pyinyaungensis in having significantly higher mean numbers of infralabial scales (IL; 8.0 vs. 6.0), fourth toe lamellae (4TL; 17.0 vs. 15.0), and having domed to weakly conical and weakly keeled body tubercles vs. raised and moderately keeled body tubercles (Tables 7 and 8). It differs from the new species (see below) from Mt. Popa and the Bago Yoma by having significantly higher mean number of infralabial scales (IL; 8.0 vs. 6.5) and a significantly lower mean number of longitudinal rows of dorsal tubercles (LRT; 13 vs. 18.6). C. meersi sp. nov. is widely separated from all other species in both the PCA and DAPC and does not fall within the plot distribution of any other species. Additionally, C. meersi sp. nov. has an uncorrected pairwise sequence divergence from them of 10–13.7% (Table 3).

Remarks. Some (Dayrat, 2005) have criticized descriptions of new species based on only a single specimen and posit that this should “never” be done because such a description cannot take into account intraspecific variation that could potentially preclude its specific recognition. We are concerned about describing a new species based on a single juvenile specimen but only because the diagnosis is incomplete, not because the diagnosis has anything to do with the reality of this specimen representing a distinct independently evolving lineage. Such a notion is false and a conflation of the species’ ontology with its epistemology (Frost & Kluge, 1994). Undoubtedly, additional specimens will expand and improve the diagnosis, however, because of safety concerns we cannot return to the type locality which is now also being converted to agriculture land. Thus, we felt it prudent to describe this species for potential protective status as soon as possible rather than delay its publication for the sake of a better diagnosis. The weak part of recognizing this specimen as a distinct species is not the incomplete diagnosis, but that the species is being delimited on the basis of a single-locus mtDNA phylogeny. It is well-documented that mtDNA phylogenies can reveal significant structure in a data set by recovering sequentially nested monophyletic groups even though within that same data set, nuclear genes can indicate significant gene flow among these groups (Shaw, 2002; Fisher-Reid & Wiens, 2011; Toews & Brelsford, 2012), thus precluding their species status. This weakens any hypothesis of specific identity based solely on mtDNA data. Nonetheless, given the current data available concerning its phylogenetic relationships and the significant and discrete morphological and color pattern differences separating C. meersi sp. nov. from its congeners, we regard the specific identity of C. meersi sp. nov. as a testable hypothesis.

Cyrtodactylus myintkyawthurai sp. nov.

Mt. Popa Bent-toed Gecko

(Fig. 8)

urn:lsid:zoobank.org:act:6B634A60-C1C5-492F-9AF8-5E2FD62D137C

Figure 8 Type specimens of Cyrtodactylus myintkyawthurai sp. nov.

Cyrtodactylus myintkyawthurai sp. nov. from the type locality of Taung Twin Chaung camp, Mt. Popa, Kyauk-pa-taung Township, Mandalay Region, Myanmar. (A) Adult male holotype LSUHC 13808. (B) Adult male paratype LSUHC 13807. (C) Subadult male paratype 13806. (D) Juvenile male paratype LSUHC 13809. Photos by L. Lee Grismer.

Cyrtodactylus fea Wood et al., 2012:995; Agarwal et al., 2014:147; Brennan et al., 2017:3.

Holotype. Adult male LSUHC 13808 collected on October 27, 2017 at 1,030 h by Evan S.H. Quah, Perry L. Wood Jr, Matthew L. Murdoch, Myint Kyaw Thura, Aung Lin, Robert E. Espinoza, Tun Oo, and L. Lee Grismer from Taung Twin Chaung camp, Mt. Popa, Kyauk-pa-taung Township, Mandalay Region, Myanmar (20.93087°N, 95.22580°E; 978 m in elevation).

Paratypes. LSUHC 13500–02, 13505, 13806–07, 13809 bear the same collection data as the holotype. CAS 245200–03 was collected on August 1, 2007 by A.K. Shein, S.W. Kyi, and J.V. Vindum from Central Bago Yoma Range, between Pallan Gyi Forest Camp and summit, Bago Region, Myamnar (18.92108°N, 95.814027°E, 268 m). CAS 222147 was collected on August 11, 2001 by H. Win and A.K. Shein from the Kyetshar Elephant Camp, Kyet Shar Village, Aok Twin Township, Bago Region, Myanmar (18.88094°N, 96.07922°E, 263 m elevation). CAS 222128 was collected on July 31, 2001 by H. Win, A.K. Shein and H. Tun from the Ka Baung Reserve, Aok Twin Township, Taung Oo District, Bago Region, Myanmar (18.834750°N, 96.18638°E, 160 m elevation).

Diagnosis. Cyrtodactylus myintkyawthurai sp. nov. differs from other species in the peguensis group by having the unique combination of six or seven supralabials and six or seven infralabials; 28–33 paravertebral tubercles; 17–23 longitudinal rows of body tubercles; 32–36 ventral scales; 17–19 subdigital lamellae on the fourth toe; 12–20 femoral pores in males; 7–9 precloacal pores in males; two rows of post-precloacal scales; raised, moderately to strongly keeled body tubercles; and a maximum SVL of 75.1 mm (Table 7).

Description of holotype. Adult male, SVL 57.6 mm; head moderate in length (HL/SVL 0.28), wide (HW/HL 0.70), somewhat flattened (HD/HL 0.43), distinct from neck, triangular in dorsal profile; lores inflated, prefrontal region concave, canthus rostralis rounded; snout elongate (ES/HL 0.40), rounded in dorsal profile; eye large (ED/HL 0.27); ear opening triangular, moderate in size (EL/HL 0.13); eye to ear distance greater than diameter of eye; rostral rectangular, partially divided dorsally, bordered posteriorly by large left and right supranasals separated an internasal, laterally by first supralabials; external nares bordered anteriorly by rostral, dorsally by large supranasal, posteriorly by two postnasals, ventrally by first supralabial; six (R,L) rectangular supralabials extending to below midpoint of eye; six (R,L) infralabials tapering smoothly to below posterior margin of orbit; scales of rostrum and lores flat, larger than granular scales on top of head and occiput; scales on top of head and occiput intermixed with slightly enlarged tubercles; dorsal supraciliaries not elongate or keeled; mental triangular, bordered laterally by first infralabials and posteriorly by large, left and right trapezoidal postmentals that contact medially for 60% of their length posterior to mental; one row of slightly enlarged chinshields extending posteriorly to third infralabial; and gular and throat scales small, granular, grading posteriorly into larger, flatter, smooth, subimbricate to imbricate, pectoral, and ventral scales.

Body relatively short (AG/SVL 0.47) with weak ventrolateral folds; dorsal scales small, interspersed with larger, moderately to strongly keeled, semi-regularly arranged keeled tubercles; tubercles extend from occiput onto base of tail but no farther; tubercles on occiput and nape smaller than those on posterior portion of body that are larger and keeled; approximately 17 longitudinal rows of dorsal tubercles; 31 paravertebral tubercles; approximately 35 flat, imbricate, ventral scales larger than dorsal scales; seven pore-bearing precloacal scales; two rows of large post-precloacal scales; and no deep precloacal groove or depression.

Forelimbs moderate in stature, relatively short (FL/SVL 0.16); flat scales of forearm larger than those on body, not interspersed with tubercles; palmar scales raised; digits slightly narrower distal to inflections; claws well-developed, sheathed by a dorsal and ventral scale; hind limbs more robust than forelimbs, moderate in length (TBL/SVL 0.15), covered dorsally by granular scales interspersed with slightly larger, weakly keeled tubercles and anteriorly by large, flat, imbricate scales; ventral scale of femora flat, imbricate, larger than dorsals, lacking a row of enlarged femoral scales; six (R,L) pore-bearing femoral scales; small postfemoral scales form an abrupt union with large, flat ventral scales of posteroventral margin of thigh; subtibial scales flat, imbricate; plantar scales raised; digits relatively short, well-developed, inflected at basal, interphalangeal joints; 17 subdigital lamellae on fourth toe; and claws well-developed, base of claw sheathed by a dorsal and ventral scale; two enlarged postcloacal tubercles at base of tail; and postcloacal scales flat.

Original tail moderate in proportions, 64.0 mm in length, 5.9 mm in width at base, tapering to a point; dorsal scales of base of tail granular rapidly becoming flatter posteriorly; and intermittent rows of transversely enlarged, median, subcaudal scales twice as wide as long, not extending onto lateral margin of tail.

Coloration in life (Fig. 8). Dorsal ground color of head body, limbs, and tail yellow; top of head bearing large, dark-brown, irregularly shaped blotches edged in yellow; dark-brown, wide, nuchal loop extending from posterior margin of one eye, across occiput, to posterior margin of other eye; nape bearing a large, dark-brown band edged in yellow; four wide, dark-brown body bands between limb insertions edged in yellow with paravertebral sections; one dark-brown post-sacral band edged in yellow bearing paravertebral sections; 10 dark-brown caudal bands wider than 11 yellow, caudal bands with darkened centers; dorsal portion of forelimbs darkly banded; dorsal portion of hind limbs bearing irregularly shaped, dark-brown blotches edges in yellow; flanks with a series of 8–10 dark-brown, round to irregularly shaped blotches of varying sizes edged in yellow. All ventral surfaces generally beige, immaculate except for ventral surfaces of forelimbs, forelegs, hands, feet, and posterior section of tail bearing dark pigment.

Variation. Variation in the paratypes of C. myintkyawthurai sp. nov. is most notable in the dorsal banding pattern. Specimens from the central Bago Yoma Range tend to have dorsal bands that are divided along the midline, thus manifesting distinct paravertebral elements (CAS 222147, 245201–03) although this is not so evident in CAS 245200 and CAS 222128 which have somewhat of an anomalous pattern with some elongate, irregularly shape blotches. The banding pattern in specimens from Mt. Popa lacks complete midline bifurcation although the dorsal bands have distinct paravertebral elements. The dorsal bands of all the paratypes except CAS 245200, 245203, and LSUHC 13505 are relatively wider than those of the holotype LSUHC 13808. Specimens CAS 245200, 245202 and LSUHC 13500–02, 13806, and 13809 have regenerated tails bearing dark mottling. Specimens CAS 222128 and LSUHC 13505 have broken tails. Adult females have dimpled scales or 0–8 femoral pores. Other meristic and mensural variation in the type series is presented in Table 6.

Distribution. C. myintkyawthurai sp. nov. ranges throughout Mt. Popa, Mandalay Region and the central section of the Bago Yoma Range, Bago Region (Fig. 2).

Etymology. The specific epithet, myintkyawthurai, is a patronym honoring Myint Kyaw Thura for his contributions to the study of herpetology in Myanmar, his discovery of several new species, and his collaboration with foreign researchers.

Natural History. At both Mt. Popa and in the central Bago Yoma Range, C. myintkyawthurai sp. nov. occurs in hilly regions covered in deciduous dipterocarp forest up to 978 m in elevation (Fig. 9). The Mt. Popa specimens were collected at night from 0.05 to 1 m above the ground on rocks, the trunks of small trees, on leaves or on the ground amongst small rocks.

Figure 9 Habitat of Cyrtodactylus myintkyawthurai sp. nov.

Mixed dry dipterocarp forest habitat of Cyrtodactylus myintkyawthurai sp. nov. near the type locality of Taung Twin Chaung camp, Mt. Popa, Kyauk-pa-taung Township, Mandalay Region, Myanmar. Photo by L. Lee Grismer.

Comparisons. See previous comparison sections.

Discussion

Members of the peguensis species group all appear to be habitat generalists within the hilly regions they inhabit. C. pyinyaungensis have been found crawling on karst boulders in dry secondary forest as well as highly disturbed, burned over lowland forests/agricultural areas. C. myintkwawthurai sp. nov. is know from dry deciduous forests and mixed dipterocarp forests and C. meersi sp. nov. and C. peguensis occur in highly disturbed lowland forests bordering agricultural fields. No natural history data were provided with the description of C. annandalei (Bauer, 2003). This lack of microhabitat preference suggests that additional populations of this species group are very likely to be found in other hilly areas both within and along the fringes of the Ayeyarwady Basin. It also indicates that agricultural practices in these areas should be monitored for purposes of conservation.

The rapidly growing awareness of the diversity of karst-associated species of Cyrtodactylus and other gekkonids in Myanmar (Grismer et al., 2017a, 2017b, 2018a, 2018b) mirrors that in other Southeast Asian nations (see discussions in Grismer et al., 2016a, 2016b, 2016c; Wood et al., 2017) but is in no way surprising given Myanmar’s vast regions of unexplored karstic habitats. What is surprising, however, is the diversity within lineages restricted to the relatively featureless Ayeyarwady Basin. The peguensis species group collectively account for at least seven species of a monophyletic group that ranges disjunctly from the eastern foothills of the Chin Hills along the western fringe of the Ayeyarwady Basin to the western foothills of the Shan Hills along the eastern fringe of the Ayeyarwady Basin (Fig. 1). Additionally, the low hills of the centrally located Bago Yoma are inhabited by C. peguensis, C. myintkyawthurai sp. nov., and C. meersi sp. nov. The allopatric distribution of these species is likely due to range fragmentation resulting from sedimentation from the numerous river courses—the Ayeyarwady and Sittaung being the largest—since at least the Lower Miocene-Upper Pliocene (Agarwal et al., 2014). The other eight species found within the Ayeyarwady Basin or along its mountainous fringes that are not part of the peguensis species (Bauer, 2003; Mahony, 2009) and are not monophyletic but nested within lineages containing Indian, Bangladeshi, and Himalayan taxa from farther west, indicating that colonization of the Ayeyarwady Basin is the result of multiple invasions since at least the Upper Miocene (Agarwal et al., 2014).

Conclusions

A phylogenetic taxonomy of species in the C. peguensis species group from the Ayeyarwady Basin of Myanmar recovers at least seven species, four of which are undescribed. Three species, including C. peguensis occur in the low hills of the Bago Yoma mountain range one of which, C. meersi sp. nov., is syntopic with C. peguensis. As more lowland hilly areas associated with the Ayeyarwady Basin are surveyed, more new species of this group are likely to be discovered. These discoveries continue the recent surge of descriptions of new species of Cyrtodactylus that are being discovered in Myanmar.

Supplemental Information

Supplemental Information 1 Nexus file.

Nexus file of raw data for the phylogenetic analyses.

Click here for additional data file.

Supplemental Information 2 Raw data for statistics.

Data file used for all statistical analyses in R.

Click here for additional data file.

We wish to thank Mr. Win Naing Thaw of the Ministry of natural Resources and Environmental Conservation Forest Department for collection and export permits. We thank the staff of the Shwe Gue Gu Hotel and Genious Coffee for their hospitality.

Additional Information and Declarations

Competing Interests

Author Contributions

Animal Ethics

Field Study Permissions

DNA Deposition

Data Availability

New Species Registration

Aung Lin is employed by Fauna and Flora International.

L. Lee Grismer conceived and designed the experiments, performed the experiments, analyzed the data, prepared figures and/or tables, authored or reviewed drafts of the paper, approved the final draft.

Perry L. Wood Jr conceived and designed the experiments, performed the experiments, authored or reviewed drafts of the paper.

Evan S.H. Quah authored or reviewed drafts of the paper.

Matthew L. Murdoch authored or reviewed drafts of the paper.

Marta S. Grismer performed the experiments, authored or reviewed drafts of the paper.

Mark W. Herr authored or reviewed drafts of the paper.

Robert E. Espinoza authored or reviewed drafts of the paper.

Rafe M. Brown contributed reagents/materials/analysis tools.

Aung Lin arranged logistics for the expedition and helped in collecting.

The following information was supplied relating to ethical approvals (i.e., approving body and any reference numbers):

Brigham Young University’s Institutional Animal Care and Use Committee (IACUC) has approved the animal use protocol for this study (protocol # 160401).

The following information was supplied relating to field study approvals (i.e., approving body and any reference numbers):

Ministry of Natural Resources and Environmental Conservation Forest Department of Myanmar provided collecting and export permits.

The following information was supplied regarding the deposition of DNA sequences:

A nexus file of the sequences is available as a Supplemental File.

The following information was supplied regarding data availability:

Dryad Doi:10.5061/dryad.tri2k786.

The following information was supplied regarding the registration of a newly described species:

Publication LSID: urn:lsid:zoobank.org:pub:80F582E5-9FE4-4A1A-AAE8-035FA0708E11

Family Group

Gekkotidae Gray, 1825 LSID:

urn:lsid:zoobank.org:act:78CFE2D5-074A-45F3-8D68-5189AB8F2F9D

Genus Group

Cyrtodactylus Gray, 1827 LSID:

urn:lsid:zoobank.org:act:AAAC5049-5EE4-47A3-AAFF-4E50495B9064

New species

Cyrtodactylus meersi LSID:

urn:lsid:zoobank.org:act:14EEE090-BD93-4A2E-9CDC-FA4C25E91D91

Cyrtodactylus myintkyawthurai LSID:

urn:lsid:zoobank.org:act:9645421A-0997-487C-B26B-2030FC42D264

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
