# Peer review of "A phylogenetic taxonomy of the Cyrtodactylus peguensis group (Reptilia: Squamata: Gekkonidae) with descriptions of two new species from Myanmar"

_PeerJ, doi:10.7717/peerj.5575_

## Round 0.1 · original submission · Major Revisions

Dear Authors,

I have received two reviews now. The first reviewer did a very thorough review, and I would like you take the reviewer’s comments and suggestions into account. In addition to the concerns raised by the first reviewer, I have concerns of my own.

My principal concerns are:

1. C. meersi sp. nov. being described only from the holotype, and this individual is a juvenile. I find this troubling. While on the phylogeny it is a distinct branch on the phylogeny, and shows at least 10% pair-wise sequence divergence from other species, the morphological data are not convincing. Not only is it impossible to know if infralabials do not vary (are fixed at 8) – infralabials do vary in C. myintkyawthurai sp. nov., or the no. longitudinal rows of tubercles is fixed at 13 (it varies broadly from 17 to 23 in C. myintkyawthurai sp. nov.), characters listed as diagnostic (no. of paravertebral tubercles = 32) are not diagnostic (no. of paravertebral tubercles = 32 in C. peguensis). PCA does not separate this species from C. myintkyawthurai sp. nov.
2. It is unclear how you make taxonomic decisions, what do you consider a lineage and why. While you state that you follow “the general lineage concept of species de Queiroz (1998, 1999), and consider allopatric populations that are morphologically and genetically diagnosable, as unique evolutionary lineages, and thus distinct species. you do not state any explicit criteria for deciding what is a lineage/species.” you do not follow this line of reasoning when considering C. myintkyawthurai sp. nov. The two populations of C. myintkyawthurai sp. nov. are both morphologically and genetically diagnosable (if the holotype is not included), but you treat them as belonging to the same species rather than belonging to two different species. Why? Also, although this is not the main focus of this study, C. oldhami is paraphyletic. Is this really the case? Could it be that the major clades/groups of Cyrtodactylus are highly structured and have very large effective population sizes? In this case populations of a species, intraspecific variation, would/could also be genetically and/or morphologically diagnosable.
3. While you provide LSID for the publication, you do not provide LSID for the newly described species (this needs to be included in the revised MS), and I suggest providing the LSID species act together with the description of the species (right after stating the new species name).
4. Do not use the names C. meersi sp. nov. and C. myintkyawthurai sp. nov. prior to their formal descriptions.
5. Table 7 – I suggest requesting from Aaron Bauer data of the three individuals of C. annandalei so that you can report the same metrics as for other species (if these data are not already available in Bauer (2003)).

Overall, I think this study has a potential to be a fine paper, but it is premature to describe C. meersi sp. nov. I realize that fieldwork and logistics probably are not easy in Mayanmar, but conservative taxonomic decisions are always preferable in the absence of strong evidence. You have a local collaborator, a coauthor of this MS, so hopefully in the near future you will have access to additional specimens needed for a robust description of this species.

Sincerely,

Tomas Hrbek

Reviewer 1 ·

Basic reporting

Grismer et al. described two new species of Cyrtodactylus, re-described one species (C. peguensis) and recategorized at species level C. zebriacus. All these findings are based on mtDNA and morphological analyses. The article is well written, and the evidence presented by the authors support the existence of these new taxa.

The evolutionary characteristic used by the authors to distinguish the new species as well as the diagnostic features and comparisons with other species are well done.

Major changes are necessary in introduction, results reporting and discussion however these changes not will compromise the validity of the findings. For example, in introduction I detected the absent of an introductory paragraph for species delimitation and the tools using (molecular, morphology). I also suggest dividing the results into two topics: 1) Phylogenetic analyses and 2) morphology and morphometric analyses. This will clarify the results and facilitate the presentation of results the the readers. In the discussion, the authors need to discuss the limitations of the study as your species delimitation is based only on mtDNA, the absence of adults in the description of C. meersi sp. nov., the main morphological characters that differentiate the species, the new categorization of C. zebraicus, and in addition, the future of the taxonomy, systematics and conservation of this species group. In addition, the authors should consider creating a plate with figures that contain the main morphological characters that differentiate the species.

Experimental design

The authors use morphological and phylogenetic characters to access the phylogenetic taxonomy of the Cyrtodactylus peguensis group.

The experimental design is complete and the evolutionary tools that were used are efficient and informative (although with limitations). As major issue here is that the species delimitation at molecular level is not clear – see my comments below.

Validity of the findings

I consider this work important to reveal the hidden species diversity within the genus.

Additional comments

Line 42= …terms of the annual rate of newly described species (reference?)
Line 72= I don’t know if “construct” is the right word
Line 73= I suggest replace “true” by Cyrtidactylus peguensis sensu stricto
Line 78 to 95= It seems that the authors need to summarize this paragraph, all the information of this paragraph are also in methodology. I suggest a paragraph where the authors explore the importance of species delimitation using multiple characters (as the authors did), and the limitations and benefits of each character (mtDNA and morphology) for taxonomic decisions.
Line 145= Could the authors specified if the p-distance were calculated between localities? Lineages? I also note that the authors do not specify the use of p-distance value to support divergence between lineages. This is not necessary but works as additional evidence of genetic lineages differentiation.
Line 151= please add the total number of specimens that you measure for analysis
Line 169= I miss a reference for morphometric measurements, who the authors followed to make these measurements?
Line 203= …each species group = which species group? Cyrtidactylus peguensis species group? Please be more specific
Line 205=… species boundaries delimited by the molecular phylogenetic= until now in methodology, the authors do no specified how they delimited species by molecular phylogenetic analyses.
Line 209= it is necessary to clarify in previous paragraph if males and females were used or only one sex for analyses. In addition, the authors need to clarify is they used only adults (that is expected).
Line 237= Results= I suggest to divide the results into two topics: 1) Phylogenetic analyses and 2) morphology and morphometric analyses. This will clarify the results and facilitate the presentation of results for the readers.
Line 269= The DAPC analysis also provide the numbers of specimens correctly assigned to each species group. see my comments on the MS.
Line 291 to 302= all these need to go to species account. see my comments on the MS.
Line 309= Again, how the species boundaries within the lineages of C. peguensis were identified (how was species delimitation carried out)? could the authors be more specifics
Line 315= please add the geographic distance of type locality.
Line 322= start with morphology, them phylogeny and geographic distribution of the group
Line 441= please add the numbers of examined specimens for morphological analyses for all the measurements and species comparison. This will allow readers to know the variation of the samples in futures studies. Or in other case leave clear the number of specimens examined for each species in methods or in a table. In addition, add “mm” after all the measurements values.
Line 440= were any statistical tests carried out? if yes, report p-values. If not, carry out statistical analyses. (1 mm of difference is not "significantly higher" by itself). the same goes for other cases
Line 481=The author need to create a fig. to illustrate this character.
Line 550= Is there no fig. showing the differentiation between this species and others in this and others morphological characters? I suggest to create one or two new fig. comparing the principal morphological characters between the species. This will clarify the species comparison.
Line 662= Major changes here. The discussion need to be improve. The authors need to discuss the limitations of study as your species delimitation using only mtDNA, the absent of adult samples in the description of C. meersi sp. nov. the main morphological characters that differentiate the species, the new categorization of C. zebriacus, and in addition, the future of the taxonomy, systematics and conservation of this species group.
Figure 2= In this fig, the authors need to select with a bar or a new color the C. peguensis specimen from type locality.

Annotated reviews are not available for download in order to protect the identity of reviewers who chose to remain anonymous.

·

Basic reporting

This manuscript is excellent and in very good state. It is well written in clear English, references are equated. I only found some minor (trivial) comments which I made directly on the PDF.
Figures are excellent and the new species are well described.
I know some member of this research group and they are very exhaustive in their studies.

Experimental design

The methodology section is very clear, and provide all the details that warrant reliability. This research was very rigorous.

Validity of the findings

I think revealing new species in this poorly explored region of Myanmar is remarkable. Myanmar is suffering from lots of social problems and I can only image the conditions the authors have to work there. However, they manage to circunvent this social problems, at least it looks like the country is very safe, and this paper might help improving the image of this region.

Additional comments

I found some minor typos in the pdf attached

---

## Round 0.2 · Minor Revisions

The authors Appealed the prior decision, and I was asked to step in and make a final adjudication.

Because the manuscript is now with me for handling, I will outline the minor adjustments I would like to see to move the manuscript forward.

1) add accession numbers to GenBank sequences in Table 1.

2) add some detail on how you selected your best-fit models of evolution in the methods section.

3) add a new paragraph in the discussion on ‘caveats’ to the study and include at least -

3a) the limitations and issues associated with describing a species from a single individual (and juvenile at that) and
3b) the limitations of a single gene phylogeny for phylogenetic inference.

I know these last two points came up in the initial review and you have argued around them. I encourage you not to argue around them again, but to simply acknowledge they are limitations to the study with appropriate references to the extensive body of literature that document these limitations. Your argument for the single individual species description is based on precedence. That’s a poor argument. Clearly, it is best to describe species on extensive population sampling. Just acknowledge this and document why this is problematic and what the implications might be. Likewise, you argued that your taxonomic designations are based on a combination of morphology, biogeography, and phylogeny, not just phylogeny. Fine. But that is not the point. The point is, of course, that a mtDNA based phylogeny without a nuclear component has issues. First, it is well documented that phylogenetic inference is much better with multiple loci. Second, because mitochondrial DNA is maternally inherited (in most organisms), there is potentially issues mtDNA showing differentiation, but nuclear DNA showing gene flow. This would be especially problematic to your new species conclusion. I’m not saying you need to sequence a bunch of nuclear genes and add them, rather simply acknowledge these limitations to the study in the paper.

I agree with you thoughts on the ‘conservative’ thing to do is not necessarily not describing this second new species. I also concur that the new species will make a working hypothesis available for further testing. However, I have also been involved in a few court cases on endangered species where the original taxonomic paper takes amazing precedence regardless of how well or poorly done it is. Adding these caveats allows readers (and lawyers!) to see what the limitations of the inferences are and what additional data might eventually substantiate or refute the hypothesis of a new species.

If you send back the paper with these adjustments, I will not send it out for further review, but will make a decision on the revision myself. I am happy for you to describe both species in the paper as long as you include in the discussion a paragraph on the caveats listed above. The paper has been transferred to me in the PeerJ system, so just submit the revision as you normally would.

Reviewer 1 ·

Basic reporting

No comments

Experimental design

No comments

Validity of the findings

No comments

Additional comments

Responses seem to be fine.

---

## Round 0.3 · accepted · Accept

Thanks for your attention to the previous comments. I especially like the paragraph addressing the one individual species description. I think it was necessary and adds the appropriate literature and acknowledgement of potential issues. Thanks.

#